# CB$_1$-receptor-mediated inhibitory LTD triggers presynaptic remodeling via protein synthesis and ubiquitination

Hannah R Monday[1], Mathieu Bourdenx[2,3], Bryen A Jordan[1,4], Pablo E Castillo[1,4]*

[1]Dominick P. Purpura Department of Neuroscience, Albert Einstein College of Medicine, Bronx, United States; [2]Department of Developmental and Molecular Biology, Albert Einstein College of Medicine, Bronx, United States; [3]Institute for Aging Studies, Albert Einstein College of Medicine, Bronx, United States; [4]Department of Psychiatry and Behavioral Sciences, Albert Einstein College of Medicine, Bronx, United States

**Abstract** Long-lasting forms of postsynaptic plasticity commonly involve protein synthesis-dependent structural changes of dendritic spines. However, the relationship between protein synthesis and presynaptic structural plasticity remains unclear. Here, we investigated structural changes in cannabinoid-receptor 1 (CB$_1$)-mediated long-term depression of inhibitory transmission (iLTD), a form of presynaptic plasticity that involves a protein-synthesis-dependent long-lasting reduction in GABA release. We found that CB$_1$-iLTD in acute rat hippocampal slices was associated with protein synthesis-dependent presynaptic structural changes. Using proteomics, we determined that CB$_1$ activation in hippocampal neurons resulted in increased ribosomal proteins and initiation factors, but decreased levels of proteins involved in regulation of the actin cytoskeleton, such as ARPC2 and WASF1/WAVE1, and presynaptic release. Moreover, while CB$_1$-iLTD increased ubiquitin/proteasome activity, ubiquitination but not proteasomal degradation was critical for structural and functional presynaptic CB$_1$-iLTD. Thus, CB$_1$-iLTD relies on both protein synthesis and ubiquitination to elicit structural changes that underlie long-term reduction of GABA release.

*For correspondence:
pablo.castillo@einsteinmed.org

Competing interests: The authors declare that no competing interests exist.

## Introduction

Synaptic plasticity, the ability of synapses to change their strength in response to activity or experience, underlies information storage in the brain. While presynaptic forms of plasticity, that is long-term synaptic strengthening (long-term potentiation or LTP) and weakening (long-term depression or LTD) due to long-lasting increase and decrease in neurotransmitter release, respectively, are widely expressed in the brain, their mechanism remains poorly understood (*Castillo, 2012*; *Monday and Castillo, 2017*; *Monday et al., 2018*; *Yang and Calakos, 2013*). A good example of a ubiquitous form of long-lasting reduction of transmitter release is type-1 cannabinoid receptor (CB$_1$)-mediated LTD (*Araque et al., 2017*; *Castillo et al., 2012*; *Heifets and Castillo, 2009*). Here, endogenous cannabinoids (eCBs) are released upon activity and travel in a retrograde manner to bind presynaptic CB$_1$, a G$_{i/o}$-coupled receptor, resulting in CB$_1$-LTD at both excitatory and inhibitory synapses. Induction of long-term eCB-mediated plasticity requires extended (minutes) CB$_1$ activation (*Chevaleyre and Castillo, 2003*; *Chevaleyre et al., 2007*; *Ronesi, 2004*). Although the presynaptic changes downstream CB$_1$ that suppress transmitter release in a long-term manner remain unclear, there is evidence that presynaptic protein synthesis is required (*Yin et al., 2006*; *Younts et al., 2016*), but what proteins are synthesized and the precise role of these proteins in CB$_1$-LTD is unclear.

Proteostatic mechanisms, the cellular processes that balance protein synthesis and degradation, are vital for neuronal function and synaptic plasticity (*Biever et al., 2019*; *Birdsall and Waites, 2019*; *Cohen and Ziv, 2019*; *Liang and Sigrist, 2018*; *Wang et al., 2017*). In postsynaptic forms of plasticity, such as NMDA receptor-dependent LTP, local protein synthesis has been tightly correlated with both consolidation of LTP and structural changes (*Bosch et al., 2014*; *Ostroff et al., 2002*; *Tanaka et al., 2008*; *Tominaga-Yoshino et al., 2008*; *Yang et al., 2008*). In particular, synthesis of β-actin, AMPA receptors, and CaMKII proteins may be critical for the increase in dendritic spine volume and synapse strength associated with LTP (*Bramham, 2008*; *Nakahata and Yasuda, 2018*; *Rangaraju et al., 2017*). Concurrent regulation of protein degradation through the proteasome and lysosome is also required for activity-dependent pre- and postsynaptic changes in synapse strength (*Biever et al., 2019*; *Cohen and Ziv, 2017*; *Hegde, 2017*; *Monday et al., 2018*). We and others have recently provided evidence for rapid (<30 min) presynaptic protein synthesis under basal conditions and during plasticity (*Hafner et al., 2019*; *Younts et al., 2016*), but whether these newly synthesized proteins participate in $CB_1$-LTD by regulating presynaptic structural changes is unknown.

Presynaptic structure and function are controlled by actin polymerization and depolymerization (*Cingolani and Goda, 2008*; *Nelson et al., 2013*). Branched actin filaments in the presynaptic compartment provide a scaffold for synaptic vesicles and the active zone (*Michel et al., 2015*; *Rust and Maritzen, 2015*). Moreover, structural changes of the presynaptic terminal are associated with altered synapse strength (*Gundelfinger and Fejtova, 2012*; *Matz et al., 2010*; *Meyer et al., 2014*; *Monday and Castillo, 2017*), and the size of the presynaptic terminal and active zone has been correlated with the postsynaptic response (*Bartol et al., 2015*; *Bourne et al., 2013*; *Meyer et al., 2014*). There is evidence that $CB_1$ activation alters the ultrastructural vesicle distribution in $CB_1$-expressing ($CB_1^+$) boutons on short time scales (*García-Morales et al., 2015*; *Ramírez-Franco et al., 2014*) and leads to retraction of growth cones in developing axons (*Roland et al., 2014*). However, it is not known whether $CB_1$-LTD is associated with morphological changes in presynaptic structure in the mature mammalian brain.

Here, to gain insights into the expression mechanisms of $CB_1$-LTD, we examined potential structural changes in $CB_1$-mediated LTD of inhibitory transmission ($CB_1$-iLTD) in the hippocampus (*Chevaleyre and Castillo, 2003*). Using high-resolution microscopy in acute rat hippocampal slices we found that this form of plasticity was associated with a reduction of presynaptic bouton volume that required protein synthesis. To test how protein synthesis could alter presynaptic structure during $CB_1$-iLTD, we used an unbiased proteomics approach to identify $CB_1$ activation-mediated changes in the proteome of hippocampal neuron cultures. $CB_1$ activation elicited an increase in proteins involved in protein synthesis, processing and degradation, whereas presynaptic and actin cytoskeletal proteins, including ARPC2 and WASF1/WAVE1 were decreased. $CB_1$-iLTD involved actin remodeling, Rac1 and the actin branching protein complex Arp2/3. Lastly, ubiquitination of proteins but not proteasomal degradation was necessary for both structural and functional $CB_1$-iLTD.

## Results

### Induction of $CB_1$-iLTD is associated with a reduction in $CB_1^+$ bouton size

Diverse forms of long-lasting synaptic plasticity require translation-dependent structural remodeling (*Bailey et al., 2015*; *Bramham, 2008*; *Nakahata and Yasuda, 2018*; *Rangaraju et al., 2017*). To test whether $CB_1$-iLTD is associated with structural plasticity, we examined changes in the morphology of $CB_1^+$ boutons following induction of $CB_1$-iLTD. To accurately measure individual bouton volume in acute hippocampal slices, we utilized high-resolution and high yield Airyscan confocal microscopy combined with 3D reconstruction (*Figure 1A*). To induce $CB_1$-iLTD the $CB_1$ agonist WIN 55,212–2 (5 µM) was bath applied for 25 min, and induction was confirmed by monitoring extracellular field inhibitory postsynaptic potentials (fIPSPs), which allows non-invasive, stable long-term assessment of inhibitory synaptic transmission (*Heifets et al., 2008*; *Younts et al., 2016*; *Figure 1B*). This LTD not only mimics synaptically induced iLTD (*Chevaleyre and Castillo, 2003*; *Chevaleyre et al., 2007*; *Heifets et al., 2008*), but also allows us to shortcut the synthesis and release of eCBs, thereby excluding potential effects of pharmacological inhibitors (see below) on these processes.

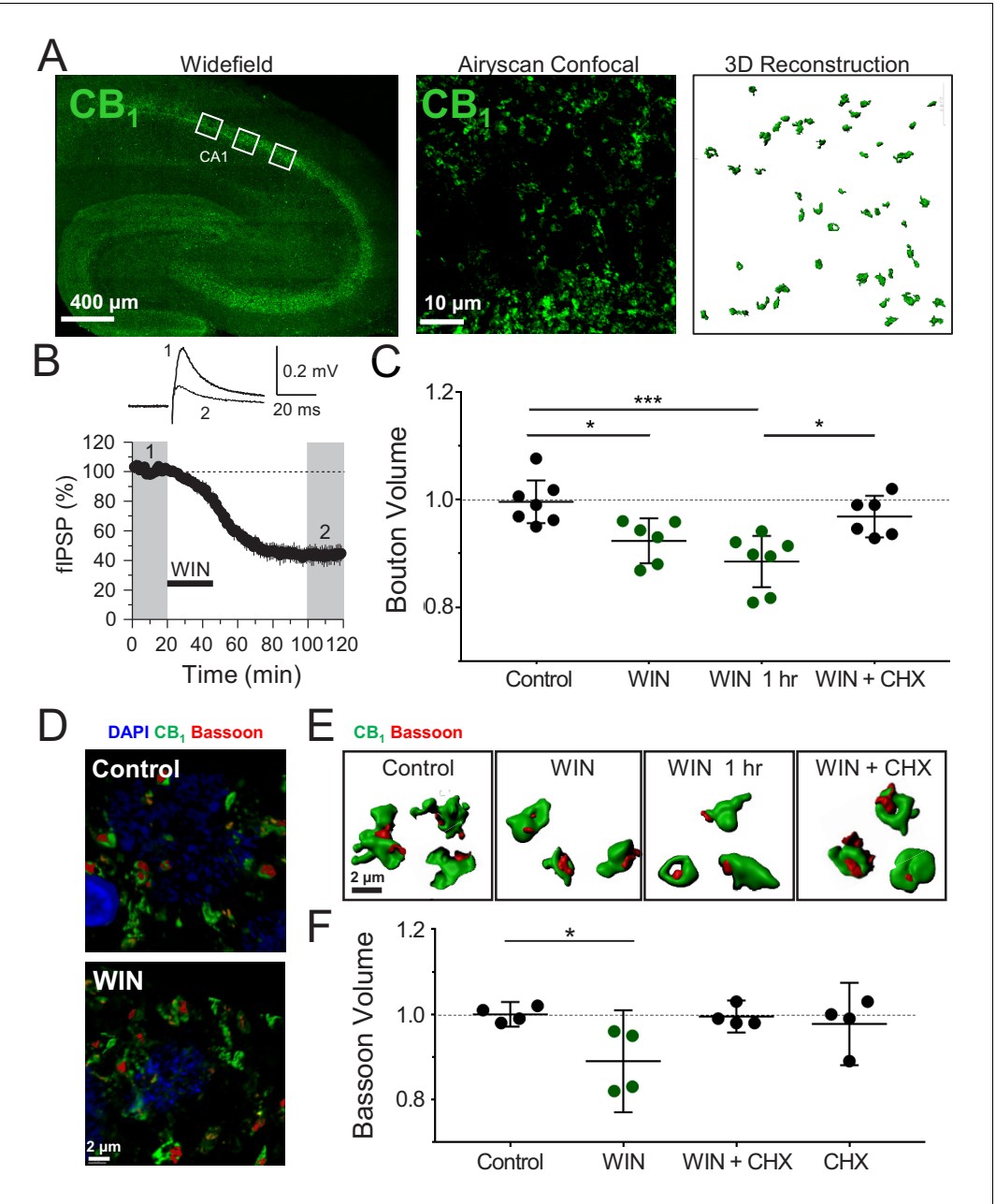

**Figure 1.** Induction of $CB_1$-iLTD is associated with a reduction in $CB_1^+$ bouton size. (**A**) *Left*, representative stitched widefield confocal of hippocampal $CB_1$ immunolabeling. White boxes indicate where high-magnification images were acquired (as seen in center panel). *Center*, High-resolution Airyscan confocal maximum projection used for 3D reconstruction of individual boutons in CA1 stratum pyramidale. *Right*, 3D reconstruction of Airyscan confocal image shown in center panel. (**B**) *Top*, representative extracellular field inhibitory postsynaptic potentials (fIPSPs) recorded in the CA1 pyramidal cell body layer in acute hippocampal slices before and after WIN treatment (5 µM, 25 min). *Bottom*, summary time-course plot showing WIN-induced depression; n = 3 slices, three animals. For all electrophysiology figures, representative traces correspond to the gray-shaded areas and in the summary time-course plots (averaged summary data expressed as normalized change from baseline ± S.E.M.). Shaded boxes in all electrophysiology figures correspond to when plasticity was analyzed with respect to baseline and when representative traces were collected and averaged. (**C**) Quantification of mean bouton volume per slice normalized to Control. Activation of $CB_1$ receptors with 5 µM WIN for 25 min led to decreased bouton volume that remained 1 hr after WIN treatment and was blocked by treatment with cycloheximide (CHX, 80 µM, applied throughout the experiment). Control: 1.0 ± 0.02 vs. WIN: 0.92 ± 0.02 vs. WIN + 1 hr: 0.89 ± 0.02 vs. CHX + WIN: 0.97 ± 0.02 (Mean ± S.E.M.); $F_{[3,22]}=8.682$; p=0.0005, one-way ANOVA with post-hoc Tukey test for multiple comparisons.

*Figure 1 continued on next page*

*Figure 1 continued*

Control vs. WIN: CI[0.005,0.14], p=0.03. Control vs. WIN 1 hr: CI[0.05,0.18], p=0.0005. WIN 1 hr vs. WIN + CHX: CI [−0.11,0.03], p=0.012. n = number of slices (three images/slice, 1–2 slices/rat, six rats/condition). For all structural plasticity figures, data are presented as points representing mean bouton volume per slice with a horizontal black line representing the mean per condition and the vertical line indicating 95% confidence interval (CI). 95% CIs are given as CI[lower CI, upper CI]. p-Values are exact. Refer to *Figure 1—figure supplement 1A* for distribution of individual bouton values. (D) Representative 3D reconstruction widefield images of Bassoon labeling inside $CB_1^+$ boutons. (E) Representative single $CB_1$ boutons with Bassoon reconstructed in 3D. (F) Quantification of mean Bassoon volume per slice normalized to Control after 25 min WIN treatment revealed a reduction in active zone volume as measured using Bassoon immunolabeling that was blocked by treatment with cycloheximide (CHX, 80 μM, applied throughout the experiment). Control: 1.0 ± 0.01 vs. WIN: 0.89 ± 0.04 vs. CHX + WIN: 1.0 ± 0.01 vs. CHX: 0.98 ± 0.03 (Mean ± S.E.M.); F[3,12]=4.11, p=0.032, one-way ANOVA with post-hoc Tukey test for multiple comparisons. Control vs. WIN: CI[0.004,0.22], p=0.042. n = number of slices (three images/slice, 1–2 slices/rat, three rats/condition). Refer to *Figure 1—figure supplement 1D* for distribution of individual Bassoon values. The online version of this article includes the following figure supplement(s) for figure 1:

**Figure supplement 1.** Individual bouton and Bassoon sizes are altered by $CB_1$-iLTD induction in a protein synthesis-dependent manner.

**Figure supplement 2.** WIN treatment affects $CB_1^+$ boutons specifically and results in reduction in VGAT volume.

Using $CB_1$ immunolabeling, which accurately approximates bouton volume (*Dudok et al., 2015*), we found that $CB_1$-iLTD is associated with a significant decrease of $CB_1$ bouton volume (*Figure 1C*). This structural change was long-lasting as it persisted for 60 min after WIN treatment (*Figure 1C*), and it was blocked by concurrent bath application with protein synthesis inhibitor cycloheximide (CHX, 80 μM), demonstrating a requirement for protein synthesis (*Figure 1C,E*, *Figure 1—figure supplement 1A*). CHX alone did not affect basal bouton volume (*Figure 1—figure supplement 1B*). This overall shrinkage by WIN treatment was driven by an increase in the proportion of the small $CB_1$ boutons and a trend toward a decrease in large boutons (*Figure 1—figure supplement 1C*). This effect was specific because volume of parvalbumin ($PV^+$) boutons in the CA1 pyramidal layer was not altered by WIN application (*Figure 1—figure supplement 2A,B*), as assessed by immuno-labeling for PV boutons (*Younts et al., 2016*), which do not express $CB_1$ receptors (*Glickfeld and Scanziani, 2006*; *Figure 1—figure supplement 2C*). As a complementary approach, we used Bassoon immunolabeling to assess the size of the presynaptic active zone. Similar to the total bouton volume, Bassoon size within $CB_1^+$ boutons was also significantly decreased following WIN application and this effect was blocked by CHX treatment (*Figure 1D–F*, *Figure 1—figure supplement 1D*). We also found that the volume of vesicular GABA transporter (VGAT), a well-established marker of inhibitory synapses, was markedly reduced in $CB_1^+$ boutons by WIN treatment (*Figure 1—figure supplement 2D*). These results strongly suggest that $CB_1$-iLTD is associated with a protein-synthesis-dependent shrinkage of $CB_1^+$ boutons, which may contribute to the long-lasting reduction in neurotransmitter release observed in this form of plasticity. Along with our previous study (*Younts et al., 2016*), our findings indicate that protein synthesis is required for both structural and functional presynaptic changes involved in $CB_1$-iLTD.

## $CB_1$ activation alters the abundance of proteins linked to protein synthesis, synaptic structure/function and energy metabolism

To glean insights into the mechanism(s) underlying structural and functional $CB_1$-iLTD, we sought to identify proteins synthesized upon $CB_1$ activation. We previously showed $CB_1$-dependent increases in protein synthesis were evident after brief $CB_1$ activation in cultured hippocampal neurons (*Younts et al., 2016*). To identify and quantitate changes in the neuronal proteome, we used Stable Isotope Labeling of Amino Acids in Cell Culture (SILAC) coupled with tandem mass spectrometry (MS/MS) (*Jordan et al., 2006*; *Zhang et al., 2011*; *Zhang et al., 2012*). Two populations of cultured rat hippocampal neurons ('medium' and 'heavy') were labeled using distinct combinations of 'medium' or 'heavy' stable-isotope weight variants of Arginine and Lysine. The two groups were treated with WIN (25 min, 5 μM) or Vehicle as before (*Figure 1*) then rapidly lysed and harvested (*Figure 2—figure supplement 1A,B*). Samples were combined and simultaneously analyzed by tandem MS/MS to identify and quantify changes induced by $CB_1$ receptor activation. To strengthen the

robustness of findings, we performed a replicate 'reverse' experiment where 'heavy' neurons were treated with WIN and observed a high degree of correlation between replicates (*Figure 2—figure supplement 1C*).

We found significant changes across the protein landscape. Examples of these proteins grouped by their suggested function are shown in *Figure 2A* (see *Supplementary file 1* for all proteins). Consistent with previous studies of axonal mRNAs, components of the protein synthesis machinery were upregulated (*Hafner et al., 2019*; *Shigeoka et al., 2016*), as well as the protein degradation machinery. A number of presynaptic proteins were downregulated following $CB_1$ activation. Notably, two key regulators of the actin cytoskeleton, Actin-related protein 2/3 complex subunit 2 (ARPC2) and Wiscott-Aldrich Associated Protein family (WASF1/WAVE1) were significantly downregulated, suggesting these proteins could be implicated in the reduction of neurotransmitter release and presynaptic volume associated with $CB_1$-iLTD. Using Gene Set Enrichment Analysis (GSEA) (*Subramanian et al., 2005*), we identified enriched functional gene ontology (GO) terms (*Figure 2B* and *Supplementary file 2*). To reduce redundancy, we clustered closely related GO terms using network analysis (*Merico et al., 2010*), where edge length corresponds to the number of overlapping proteins in the GO term, node size indicates the number of proteins belonging to the term, and color represents the enrichment score (*Figure 2C* and *Supplementary file 2*). In accordance with translational upregulation following $CB_1$ activation (*Younts et al., 2016*), the top cluster represented upregulated GO terms related to 'Protein synthesis and processing' (*Figure 2C*). The second cluster was composed of GO terms relating to 'Neuronal projections', suggestive of the structural change associated with $CB_1$-iLTD. The third cluster was GO terms associated with 'Energy metabolism' which may be representative of the previously reported $CB_1$-mediated decrease in cellular respiration (*Hebert-Chatelain et al., 2016*; *Mendizabal-Zubiaga et al., 2016*; *Figure 2C*). Examples of GO terms identified in each cluster are provided in *Figure 2—figure supplement 2*. Ingenuity pathway analysis (IPA) also identified pathways related to the processes outlined above, including eIF2 signaling, mitochondrial dysfunction, and actin cytoskeleton signaling (*Figure 2—figure supplement 1D*). Similarly, analyses using SynGO (*Koopmans et al., 2019*), an expert-curated tool to identify GO terms associated with synaptic function, linked our results to regulation of synaptic protein synthesis (*Figure 2—figure supplement 1E*). Among the differentially expressed proteins, we found that 43 proteins were upregulated and 56 proteins were downregulated by $CB_1$ activation (*Figure 2D*). Together, these results suggest that both protein synthesis and coincident degradation of structural and presynaptic proteins occur downstream of $CB_1$ activation, and could therefore be implicated in $CB_1$-iLTD.

## $CB_1$-iLTD involves actin remodeling via Rac1 and Arp2/3

$CB_1$ directly interacts with Rac1 and members of the WAVE regulatory complex (WRC), which includes actin branching modulators WASF1/WAVE1 and Arp2/3 (*Njoo et al., 2015*), and these proteins are downregulated in hippocampal neurons following $CB_1$ activation (*Figure 2A*). Therefore, regulation of the abundance of these proteins may represent a mechanism underlying structural and functional presynaptic changes involved in $CB_1$-iLTD. For example, $CB_1$ activation could reduce the presynaptic terminal volume by favoring actin depolymerization. To test this possibility, we first examined whether actin cytoskeletal dynamics were required for $CB_1$-iLTD induced structural plasticity (*Figure 1*). Using the same high-resolution microscopy and 3D reconstruction as *Figure 1*, we activated $CB_1$ in the presence of jasplakinolide (JSK, 250 nM), a reagent that promotes actin polymerization (*Holzinger, 2009*). We found that JSK application blocked the WIN-induced decrease in presynaptic bouton volume (*Figure 3A*, *Figure 3—figure supplement 1A*). These results indicate that actin dynamics likely underlie the structural changes following $CB_1$ activation. We next tested the functional requirement for actin remodeling in $CB_1$-iLTD (as in *Figure 1*). Similar to the effects on structural plasticity, bath application of JSK impaired $CB_1$-iLTD (*Figure 3B*), whereas JSK application alone had no effect on basal synaptic transmission (*Figure 3—figure supplement 1B*). These results strongly suggest that actin remodeling is critical for structural and functional $CB_1$-iLTD.

The Rac1 GTPase is one of the principal regulators of actin polymerization via WASF1/WAVE1 and Arp2/3 activity (*Derivery and Gautreau, 2010*; *Stradal and Scita, 2006*). To test the role of this pathway, we inhibited Rac1 activity using NSC 23766 (NSC), an inhibitor of Rac1-GEF interaction (*Gao et al., 2004*). $CB_1$-iLTD was impaired by application of NSC (30 μM, 25 min) during induction (*Figure 3C*). NSC alone transiently suppressed inhibitory transmission (*Figure 3D*), unlike excitatory

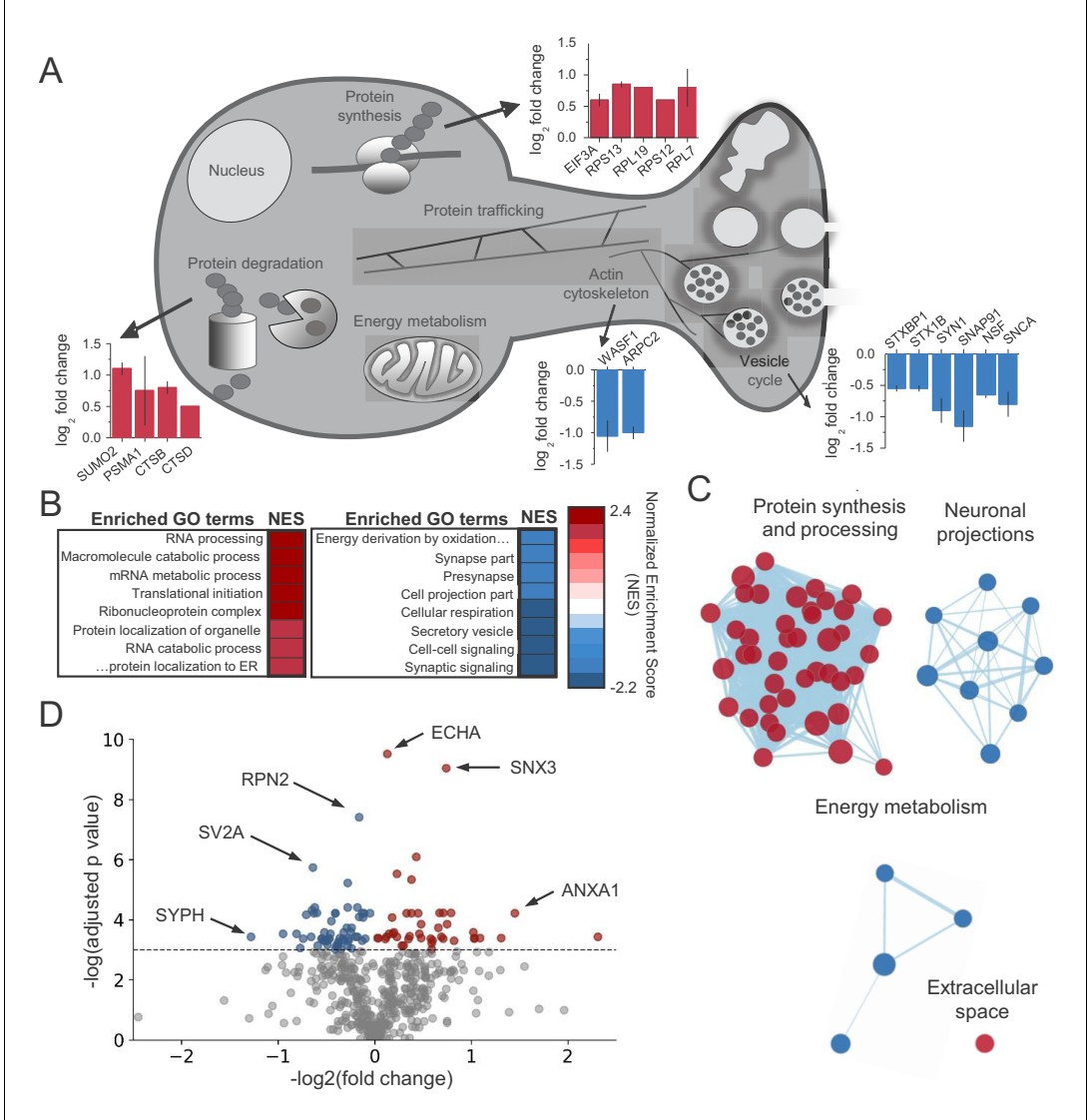

**Figure 2.** CB1 activation alters the abundance of proteins linked to protein synthesis, synaptic structure/function and energy metabolism. (**A**) Examples of proteins that were identified in enriched GO terms and were significantly altered by CB1 activation (p<0.05). Proteins are grouped by proposed biological function and average log2 fold change is plotted. (**B**) List of enriched GO terms and normalized enrichment scores (NES) as identified by GSEA. Positive NES reflects overall upregulation of proteins associated with the GO term whereas negative values indicate the opposite. (**C**) Cluster analysis of enriched/depleted GO terms from GSEA revealed four distinct biological processes that were consistently up- or downregulated by CB1 activation. Each node represents a single GO term. Node size represents magnitude of enrichment and edge length gives degree of overlap between 2 GO terms. Color represents up (red) or downregulation (blue) of proteins associated with that GO term. See *Figure 2—figure supplement 2* for examples. (**D**) Volcano plot of differentially expressed proteins between vehicle and CB1 activation. Red dots: differentially expressed proteins showing upregulation (adj. p<0.05 and log2 fold change >0). Blue dots: differentially expressed proteins showing downregulation (adj. p<0.05 and log2 fold change <0). six select top hits are highlighted: ECHA: trifunctional enzyme subunit alpha (mitochondrial); SNX3: Sorting nexin-3; RPN2: Subunit of the oligosaccharyl transferase; SV2A: Synaptic vesicle glycoprotein 2A; ANXA1: Annexin A1; SYPH: Synaptophysin.

The online version of this article includes the following figure supplement(s) for figure 2:

**Figure supplement 1.** SILAC experiment design and additional analysis.

**Figure supplement 2.** Examples of GO terms in each cluster are provided.

transmission (*Hou et al., 2014*), and this effect was associated with a decrease in PPR (*Figure 3—figure supplement 1C*), suggesting Rac1 activity regulates GABA release. To directly test the role of Arp2/3 in CB1-iLTD we utilized CK-666 (100 µM), a compound that inhibits Arp2/3-mediated actin assembly by stabilizing the inactive conformation of Arp (see *Figure 3G*; *Basu et al., 2016*;

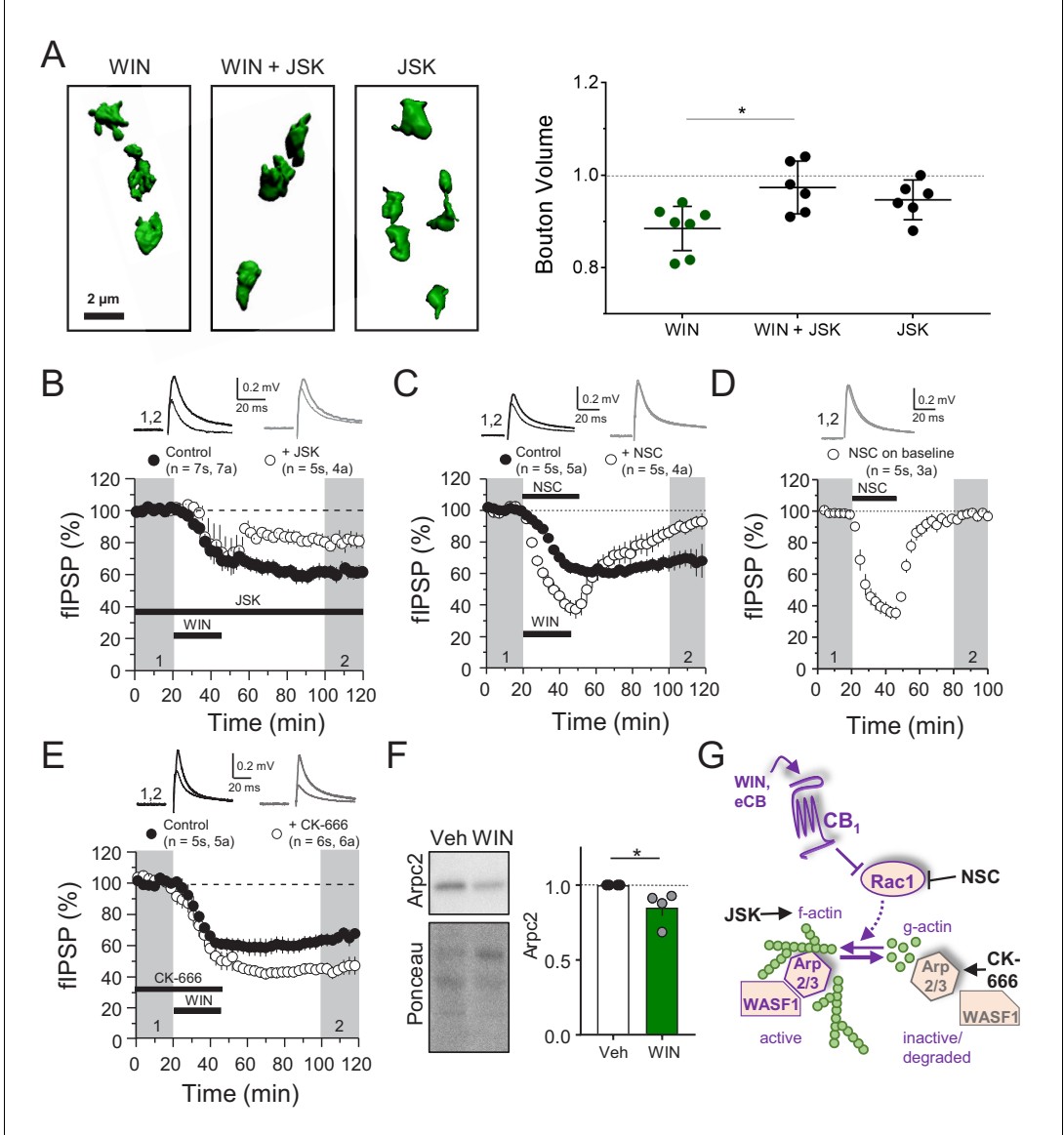

**Figure 3.** CB$_1$-iLTD involves actin remodeling via Rac1 and Arp2/3. (**A**) *Left*, representative single boutons reconstructed in 3D. *Right*, quantification of mean bouton volume per slice normalized to control. Activation of CB$_1$ receptors with WIN for 25 min led to decreased bouton volume that was blocked by treatment with jasplakinolide (JSK, 250 nM). Summary data expressed as normalized change from Control. WIN + 1 hr: 0.89 ± 0.02 vs. WIN + JSK: 0.97 ± 0.02 vs. JSK: 0.95 ± 0.02 (Mean ± S.E.M.); F[2,16]=5.56, p=0.015, one-way ANOVA with post-hoc Tukey test for multiple comparisons. WIN vs. WIN + JSK: CI[−0.16,–0.018], p=0.014. n = number of slices (three images/slice, 1–2 slices/rat, six rats/condition). For all structural plasticity figures, data are presented as points representing mean bouton volume per slice with a horizontal black line representing the mean per condition and the vertical line indicating 95% confidence interval (CI). 95% CIs are given as CI[lower CI, upper CI]. p-Values are exact. Refer to *Figure 3—figure supplement 1A* for distribution of individual bouton values. (**B**) CB$_1$-iLTD is impaired by bath application of actin-stabilizing drug, jasplakinolide (JSK, 250 nM). Extracellular field inhibitory postsynaptic potential (fIPSP) were recorded in the CA1 pyramidal cell body layer in acute hippocampal slices. Control: 61.4 ± 4% vs. JSK: 80.5 ± 4%; p<0.05, unpaired t-test. Unless otherwise specified, n = number of slices (**s**), number of animals (**a**). (**C**) CB$_1$-iLTD was blocked by acute bath application of the Rac1 inhibitor NSC (30 μM). Control: 68.8 ± 6% vs. NSC23766: 89.7 ± 4%; p<0.05, unpaired t-test. (**D**) NSC (30 μM) bath application reversibly depressed basal transmission. NSC: 98 ± 2%, one sample t-test, p>0.05. (**E**) CB$_1$-iLTD is enhanced by acute bath application of the Arp2/3 inhibitor CK-666 (100 μM). Control: 63.8 ± 4% vs. CK-666: 45.2 ± 4%, p>0.05, unpaired t-test. (**F**) *Left*, representative Western blots of staining for Arpc2 and Ponceau loading control in vehicle or WIN-treated hippocampal cultures. *Right*, Arpc2 was downregulated in hippocampal neuron cultures after CB$_1$ activation with WIN (5 μM, 25 min). Arpc2 (Fold of Veh): 0.851 ± 0.06, U = 16, Z = 2.31, * indicates p<0.05, Mann-Whitney. Dots represent individual values for four independent experiments. Data in the bar plot represent mean ± S.E.M. (**G**) Proposed model of CB$_1$-iLTD pathway and mechanism of action of pharmacological reagents. CB$_1$ activation triggers protein synthesis (not shown) and leads to inhibition of Rac1 which causes disassembly of the Arp2/3-WASF1 complex. Arp2/3 is degraded leading to actin remodeling. Actin dynamics are

*Figure 3 continued on next page*

*Figure 3 continued*

required for CB$_1$-iLTD. NSC inhibits Rac1-GEF interaction. CK-666 stabilizes the inactive conformation of Arp2/3, preventing it from binding actin filaments. JSK stabilizes actin filaments and promotes polymerization.

The online version of this article includes the following figure supplement(s) for figure 3:

**Figure supplement 1.** Individual bouton sizes are altered by CB$_1$-iLTD and dependent on actin dynamics but actin inhibitors have no effect on basal transmission.

*Hetrick et al., 2013*). CK-666 bath application enhanced CB$_1$-iLTD (*Figure 3E*) suggesting that Arp2/3 participates in CB$_1$-iLTD. Unlike NSC, CK-666 had no effect on basal inhibitory transmission (*Figure 3—figure supplement 1D*), presumably because the inhibitor stabilizes the inactive (unbound) Arp2/3, but does not affect the Arp2/3 bound to actin filaments. ARPC2 protein, an essential component of the Arp2/3 complex, was reduced upon CB$_1$ activation in hippocampal neuron cultures (*Figure 3F*). We speculate that during normal CB$_1$-iLTD, CB$_1$ activation-mediated Rac1 inhibition leads to removal of Arp2/3 from actin branches, and the subsequent degradation of Arp2/3 (*Figure 3G*). The enhancement of CB$_1$-iLTD by CK-666 application probably occurs because the unbound Arp2/3 that has not been degraded following CB$_1$ activation becomes inhibited and cannot maintain actin branches, thereby resulting in further depolymerization. Together, our findings suggest that Rac1 signaling and loss of Arp2/3 likely underlie the actin remodeling required for functional and structural CB$_1$-iLTD (*Figure 3G*).

## CB$_1$-iLTD requires ubiquitination, but not degradation by the proteasome

The simplest interpretation of our findings is that CB$_1$-induced degradation of ARPC2 and WASF1/WAVE1 led to impaired actin remodeling and reduced presynaptic bouton size (see *Figure 2*). Congruent with this idea, presynaptic release machinery and cytomatrix proteins were consistently downregulated (*Figure 4A*), whereas proteins involved in the ubiquitin/proteasome system (UPS) were upregulated (*Figure 4B*). We confirmed that presynaptic proteins identified in the SILAC screen, Munc18-1, Synapsin-1, and α-Synuclein, were significantly reduced by WIN (25 min, 5 µM) in hippocampal cultures (*Figure 4C*), suggesting rapid protein degradation upon CB$_1$ activation. To test whether presynaptic proteins are downregulated locally in acute hippocampal slices, we prevented anterograde and retrograde axonal transport by incubating slices in nocodazole (1 hr, 20 µM), an agent that depolymerizes axonal microtubules (*Barnes et al., 2010*; *Younts et al., 2016*). We found that CB$_1$ activation with WIN reduced Synapsin-1 puncta intensity in CB$_1^+$ boutons despite blockade of axonal transport (*Figure 4D*), as measured by immunostaining and quantitative Airyscan microscopy, consistent with local downregulation. These results suggest that CB$_1$ activation elicits rapid downregulation of presynaptic proteins in culture and in acute slices which likely contributes to the reduction in GABA release associated with CB$_1$-iLTD and may be mediated by degradation by the UPS.

Next, we assessed the overall contribution of the UPS to CB$_1$-iLTD. First, to dynamically assess the UPS pathway, we measured K48-linked ubiquitinated proteins, the canonical form of ubiquitin linkage (*Dantuma and Bott, 2014*), following induction of CB$_1$-iLTD in acute rat hippocampal slices in presence or absence of the specific proteasome inhibitor, MG-132. We found that both net flux, that is the amount of ubiquitinated proteins degraded by the proteasome (difference between UbK48 level when proteasomal degradation is blocked and UbK48 level under normal conditions), and the rate of degradation, (measured by the ratio of UbK48 levels between blocked and basal conditions), were significantly increased. These results suggest both a larger pool of protein to degrade as well as a faster turnover rate (*Figure 5A*). However, to our surprise, CB$_1$-iLTD was unaffected by application of the proteasomal inhibitor MG-132 (5 µM) during the baseline and induction (*Figure 5B*). MG-132 alone had no lasting effect on basal transmission either (*Figure 5—figure supplement 1A*). As a positive control, MG-132 application in interleaved slices resulted in accumulation of ubiquitinated proteins (*Figure 5—figure supplement 1B*). Therefore, while UPS activity is increased downstream of CB$_1$ activation, proteasomal degradation is not necessary for CB$_1$-iLTD.

Ubiquitination not only targets proteins for degradation, but can also affect their localization and function (*Hamilton and Zito, 2013*). We analyzed ubiquitination sites on a subset of proteins that

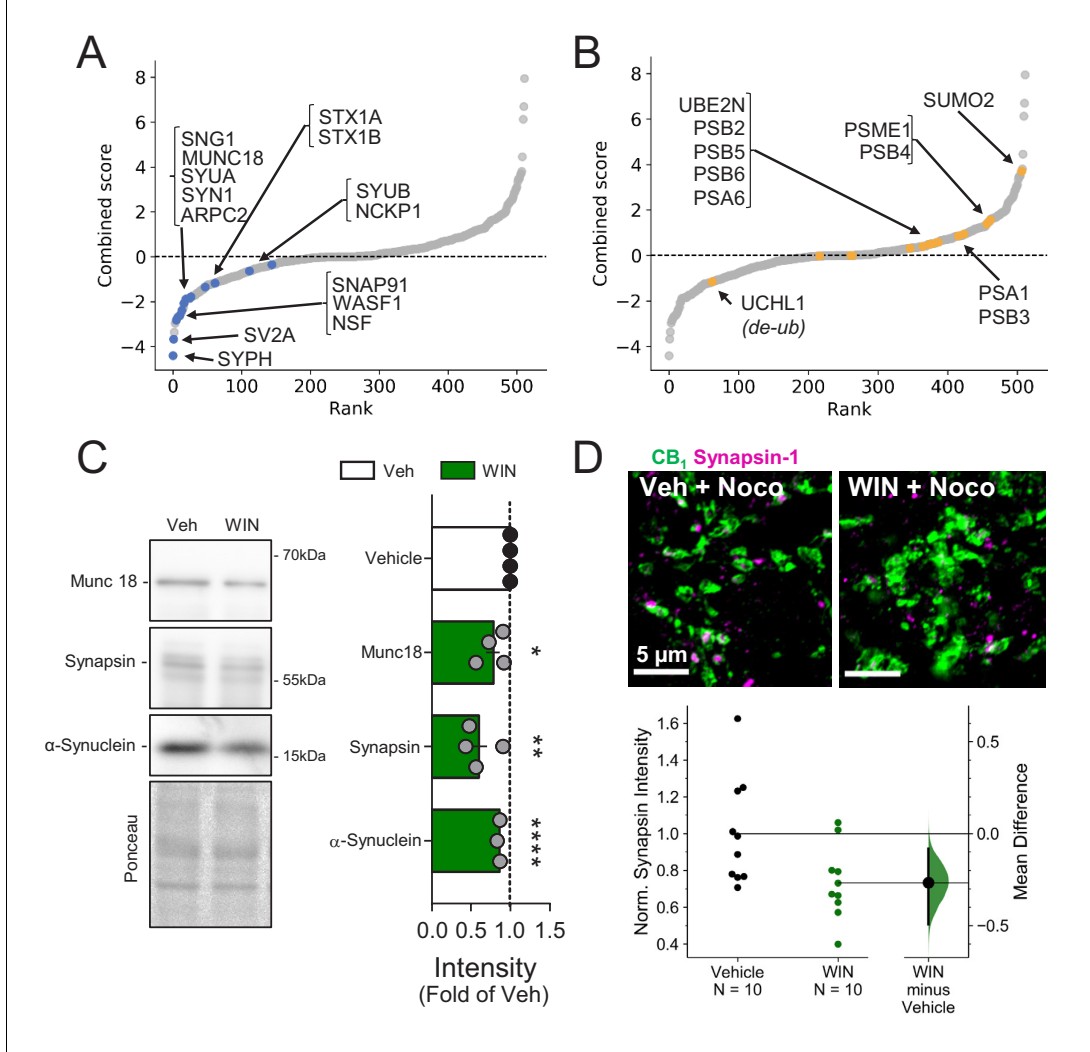

**Figure 4.** Presynaptic proteins are rapidly reduced following CB$_1$ activation. (**A**) Combined score plot of differentially expressed proteins between vehicle vs. CB$_1$ activation with WIN. Blue dots are manually selected presynaptic cytomatrix and release machinery proteins. SYPH: Synaptophysin; SV2A: Synaptic vesicle glycoprotein 2A; Synaptosome associated protein 91: SNAP91; WASF1: Wiskott-Aldrich syndrome protein family member 1; NSF: Vesicle-fusing ATPase; SNG1: synaptogyrin; MUNC18: mammalian homologue of UNC-18; SYUA: alpha-synuclein; SYN1: Synapsin-1; ARPC2: Arp complex subunit 2; STX1A-B: Syntaxin1A-1B; SYUB: beta-synuclein; NCKP1: Nck-associated protein 1. (**B**) Combined score plot of differentially expressed proteins between vehicle vs. CB$_1$ activation with WIN. Yellow dots are manually selected ubibquitin-proteasome system-related proteins. Note that the only downregulated protein is a de-ubiquitinase. UCHL1: Ubiquitin carboxyl-terminal hydrolase isozyme L1; PSB 2,3,4,5,6: Proteasome subunit beta type-2,4,5,6; PSA 1,6: Proteasome subunit alpha type 1,6; SUMO2: Small ubiquitin-related modifier 2; PSME1: Proteasome activator complex subunit 1. (**C**) *Left*, representative Western blot images of staining for presynaptic proteins Munc18-1, Synapsin-1, and α-Synuclein and Ponceau loading control in vehicle vs. WIN-treated hippocampal cultures (5 µM, 25 min). *Right*, quantification of three experimental replicates normalized to Vehicle revealed a decrease in all three proteins consistent with SILAC. Munc18-1: 0.78 ± 0.09, p<0.05; Synapsin-1: 0.60 ± 0.11, p<0.01; α-Synuclein: 0.86 ± 0.01, p<0.0001, unpaired t-test, n = number of cultures. (**D**) *Top*, Airyscan confocal representative images of CB$_1^+$ boutons in acute hippocampal slices in CA1 pyramidal layer showing colocalization of CB$_1^+$ boutons (green) and Synapsin-1 (magenta). *Bottom*, Gardner-Altman estimation plot showing the mean difference between Control and WIN of intensity of Synapsin-1 puncta within CB$_1^+$ boutons per slice was significantly diminished by WIN application (5 µM, 25 min). Both groups are plotted on the left axes; the mean difference is plotted on a floating axes on the right as a bootstrap sampling distribution. The mean difference is depicted as a dot; the 95% confidence interval is indicated by the ends of the vertical error bar. Control: 1.0 ± 0.09, WIN: 0.73 ± 0.06 (Mean ± S.E.M.), n = number of slices (10 slices, four rats/

*Figure 4 continued on next page*

*Figure 4 continued*

condition). The unpaired mean difference between Control and WIN is −0.267 [95.0%CI −0.494, −0.0819]. Two-sided permutation t-test, p=0.0234.

were decreased by $CB_1$ activation and found that most ubiquitination sites (~60%) were located in protein-protein or protein-membrane interaction domains (*Figure 5—figure supplement 1C*; *Akimov et al., 2018*), indicating that ubiquitination of these proteins could impact their function. We hypothesized that perhaps protein ubiquitination itself, independent of degradation, may play a role in $CB_1$-iLTD. Using two structurally and mechanistically distinct E1 Ubiquitin ligase inhibitors, ziram and PYR-41 (*Rinetti and Schweizer, 2010*), we directly tested whether ubiquitination was required for $CB_1$-iLTD and found that bath application of ziram or PYR-41 blocked $CB_1$-iLTD (*Figure 5C*) but had no significant effect on basal transmission (*Figure 5—figure supplement 1D*). Moreover, inhibition of ubiquitination also blocked the $CB_1$-mediated decrease in $CB_1^+$ bouton volume (*Figure 5D*, *Figure 5—figure supplement 1E*). In summary, $CB_1$-iLTD leads to increases in UPS proteins and is associated with functional increases in proteasomal activity (*Figure 5E*). Ubiquitination is required for protein synthesis-dependent structural and functional changes of $CB_1^+$ boutons. $CB_1$-iLTD is associated with decreases in presynaptic and cytomatrix proteins, including ARPC2 and WASF1/WAVE1, and relies on actin dynamics. However, while proteasomal activity increases by $CB_1$ activation, only protein ubiquitination is required for structural and functional $CB_1$-iLTD.

## Discussion

We discovered that $CB_1$-iLTD involves structural changes of the presynaptic bouton that require protein synthesis. We identified the proteins that are up- and downregulated following $CB_1$ activation. Increased proteins are implicated in protein synthesis, processing and degradation, whereas decreased proteins are implicated in presynaptic structure, including ARPC2 and WASF1/WAVE, and function. $CB_1$-iLTD involved actin remodeling, Rac1 and Arp2/3 signaling. Unexpectedly, we found that protein ubiquitination, but not proteasomal degradation, is responsible for structural and functional $CB_1$-iLTD. Together, these findings point to a mechanism by which inhibitory presynapses can control their strength in response to $CB_1$ activation via rapid proteostatic regulation of presynaptic structural change.

### Presynaptic structural changes in $CB_1$-iLTD

While structural changes are part and parcel of postsynaptic forms of plasticity (*Bramham, 2008*; *Nakahata and Yasuda, 2018*), and changes associated with plasticity are thought to be coordinated across the synaptic cleft, the involvement of structural changes of the presynaptic terminal in forms of long-term presynaptic plasticity are less clear. Here, we provide evidence for long-term structural changes at mature $CB_1^+$ terminals associated with $CB_1$-iLTD. Previous work showed $CB_1$ receptor activation can trigger ultrastructural changes in vesicle distribution associated with short-term $CB_1$-mediated plasticity (*García-Morales et al., 2015*; *Ramírez-Franco et al., 2014*), collapse of axonal growth cones (*Berghuis et al., 2007*), and inhibitory bouton formation in response to strong postsynaptic excitation (*Hu et al., 2019*). Our data show that transient activation of $CB_1$ receptors leads to a long-term reduction of the presynaptic $CB_1^+$ compartment volume in somatic synapses onto CA1 pyramidal cells. Our findings (*Figure 3*) are consistent with a previous study showing that $CB_1$ receptors regulate actin dynamics in growth cones by directly interacting with Rac1 (*Njoo et al., 2015*), a Rho GTPase (*Mattheus et al., 2016*). By directly binding $CB_1$, Rac1 can localize the WRC which consists of WASF1/WAVE1, Cyfip1, Nap1, Abi and HSP300, at the plasma membrane (*Chen et al., 2010*; *Eden et al., 2002*). The WRC is intrinsically inactive at rest (*Derivery et al., 2009*), but upon recruitment to the membrane by Rac1 the cytoplasmic side is opened for binding to Arp2/3 and actin (*Chen et al., 2010*; *Eden et al., 2002*), leading to dissociation of WASF1. Although the exact mechanism of degradation has not been shown for Arp2/3 or WASF1/WAVE1, the non-neuronal, structurally homologous isoform WAVE2 was demonstrated to undergo activation-dependent dissociation from the WRC ubiquitination and proteasomal degradation (*Joseph et al., 2017*). Therefore, Rac1 could be required for the degradation of WASF1/WAVE1,

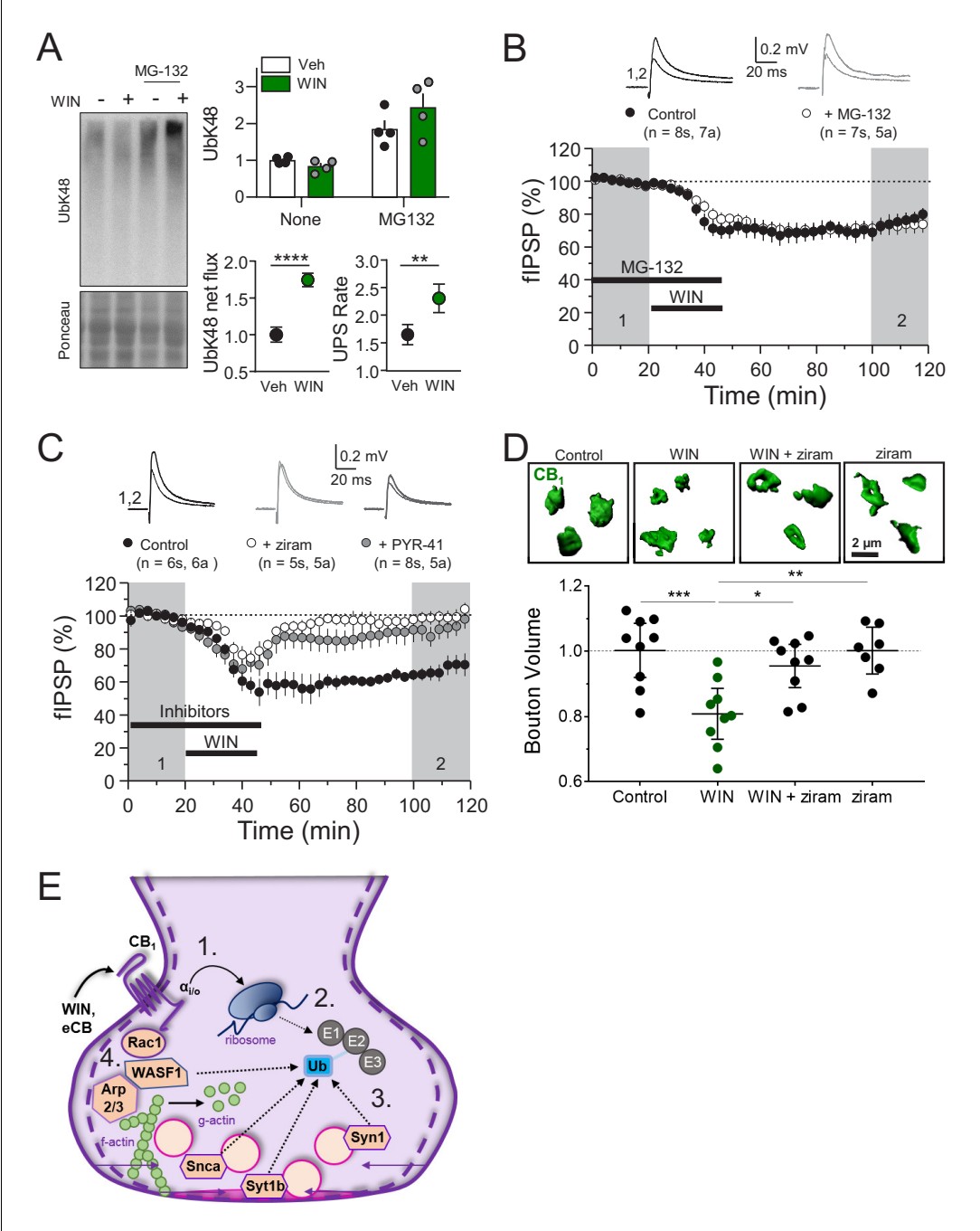

**Figure 5.** CB₁-iLTD requires ubiquitination, but not degradation by the proteasome. (**A**) *Left*, representative Western blot images of K48 polyubiquitin in hippocampal slices treated with Veh v. WIN or Veh v. WIN and MG-132. *Right, top*: Levels of K48 polyubiquitin following CB₁ activation with WIN (5 μM, 25 min). *Bottom left:* UPS net flux [difference between basal (none) and proteasome blockade (MG-132) condition] is significantly increased upon CB₁ activation. Control: 1.0 ± 0.04 vs. WIN: 1.74 ± 0.04, unpaired t-test, ****=p < 0.0001, n = 5 animals. *Bottom right*: UPS rate [ratio between MG-132 and basal condition] of K48 polyubiquitinated protein degradation is significantly increased after WIN. Control: 1.65 ± 0.08 vs. WIN: 2.31 ± 0.12, unpaired t-test, **=p < 0.01, n = 5 animals. (**B**) Blockade of the proteasome by bath application of MG-132 (5 μM) had no effect on iLTD. Control: 77.6 ± 4% vs. MG-132: 73.3 ± 5%; p>0.05, unpaired t-test. For all electrophysiology figures, averaged summary data expressed as normalized change from baseline ± S.E.M. and n = number of slices (s), number of animals (a). (**C**) Inhibiting ubiquitination with ziram (25 μM) or PYR-41 (50 μM) fully blocked iLTD. Control: 66 ± 5 vs. ziram: 99 ± 4 vs. PYR-41: 93 ± 6; F[2,19]=10.22; p<0.05, one-way ANOVA. (**D**) Blockade of E1 ubiquitin ligase function with ziram (25 μM, 25 min) rescued the volumetric decrease associated with CB₁ activation by WIN (5 μM, 25 min). *Top*, representative inhibitory boutons immunolabeled with CB₁ and reconstructed in 3D. *Bottom*, quantification of normalized mean CB₁ bouton volume per slice. Control: 1.0 ± 0.04 vs. WIN: 0.81 ± 0.03 vs. WIN + ziram: 0.95 ± 0.03 vs. ziram: 1.0 ± 0.03 (mean ± S.E.M.); F[3,30]=8.11, p=0.0004, one-way ANOVA with post-hoc Tukey test for

*Figure 5 continued on next page*

Figure 5 continued

multiple comparisons. Control vs. WIN: CI[0.07,0.32], p=0.0008; WIN vs. WIN + ziram: CI[−0.27,–0.025], p=0.014; WIN vs. ziram: CI[−0.32,–0.063], p=0.0018. n = number of slices (three images/slice, one slices/rat, nine rats/condition for Control, WIN, WIN + ziram, seven rats/condition for ziram). For all structural plasticity figures, data are presented as points representing mean bouton volume per slice with a horizontal black line representing the mean per condition and the vertical line indicating 95% confidence interval (CI). 95% CIs are given as CI[lower CI, upper CI]. p-Values are exact. Refer to *Figure 5—figure supplement 1E* for distribution of individual bouton values. (E) Schematic summary of proposed mechanism of $CB_1$–iLTD 1. $CB_1$ activation rapidly engages presynaptic protein synthesis (*Younts et al., 2016*). 2. $CB_1$–iLTD increases components of the ubiquitin/proteasome system and leads to enhanced proteasomal activity (*Figure 4A,B* and (A)) 3. Ubiquitination is required for concomitant decrease in the volume of the bouton (D) and reduction in neurotransmitter release (C). 4. $CB_1$–iLTD involves signaling via Rac1 and Arp2/3 and actin dynamics.

The online version of this article includes the following figure supplement(s) for figure 5:

**Figure supplement 1.** CB1-iLTD results in changes in proteins with ubiquitination sites in functional domains.

but its direct involvement in presynaptic degradation of Arp2/3 is not as clear. Our strategy of bath application of pharmacological inhibitors does not allow us to rule out an extra-presynaptic effect.

Other studies have suggested alternative signaling pathways by which structural changes occur downstream of $CB_1$ activation. For instance, atypical coupling of $CB_1$ to $G_{12/13}$ proteins reportedly engages Rho-associated kinase (ROCK) signaling to the actomyosin cytoskeleton (*Roland et al., 2014*), whereas in another example, β-integrin signaling to ROCK has been implicated in a cannabinoid-mediated form of LTP (*Wang et al., 2018a*). Therefore, the signaling pathways and structural changes involved downstream of $CB_1$ receptors seem to be synapse and cell-type-dependent (*Steindel et al., 2013*). Whether CB-LTD at other synapses shares similar mechanisms remains unclear.

We showed that $CB_1$-iLTD involves a rapid reduction in the size of $CB_1$ boutons and the volume of active zones that requires both actin dynamics and protein synthesis (*Figure 1*). $CB_1$ bouton size was measured by immunolabeling the $CB_1$ receptor, which, due to the extremely high density and of $CB_1$ receptors expressed on presynaptic boutons in the CA1 region of the hippocampus, has been shown to be a reliable approximation of bouton volume (*Dudok et al., 2015*). Although studies in cultured neurons suggest $CB_1$ internalization occurs (*Coutts et al., 2001*; *Hsieh et al., 1999*; *Jin et al., 1999*; *Tappe-Theodor et al., 2007*), little is known about the internalization of $CB_1$ receptor in the intact brain, which may require longer timescales than the brief $CB_1$ activation (25 min) we have used here (*Dudok et al., 2015*; *Thibault et al., 2013*). It is unlikely that $CB_1$ internalization can account for the volumetric change since the high density of $CB_1$ means that a large proportion of receptors would have to be internalized to affect measurement of the bouton size (*Thibault et al., 2013*). Moreover, the $CB_1$-mediated presynaptic shrinkage is blocked by protein synthesis inhibition whereas internalization is not protein synthesis-dependent (*Hsieh et al., 1999*) and is also detectable at the sub-presynaptic compartment level with Bassoon and VGAT labeling (*Figure 1*; *Figure 1—figure supplements 1* and *2*), two well-established markers of presynaptic structure. Intriguingly, VGAT size showed a marked reduction in $CB_{1+}$ boutons, which could reflect synaptic vesicle redistribution, consistent with previous reports showing short-term $CB_1$ activation-induced vesicle redistribution and changes in numbers of docked/primed vesicles (*García-Morales et al., 2015*; *Ramírez-Franco et al., 2014*). While the exact functional implications of presynaptic bouton and active zone shrinkage are yet unclear, associated changes in neurotransmitter release may result from altered $Ca^{2+}$ channel coupling distance with the active zone (*Nakamura et al., 2015*), less area for vesicular release, shift from multi- to univesicular release mode (*Aubrey et al., 2017*), or reorganization of transsynaptic nanocolumns (*Chen et al., 2018*; *Glebov et al., 2017*).

## Protein synthesis in presynaptic $CB_1$-iLTD likely regulates diverse cellular processes

We have recently reported that protein synthesis is required for $CB_1$-iLTD (*Younts et al., 2016*). Using a well-established unbiased proteomics approach in primary hippocampal neuron cultures (*Figure 2*; *Jordan et al., 2006*; *Zhang et al., 2011*; *Zhang et al., 2012*), we identified what proteins are synthesized upon $CB_1$ activation and thus can mediate $CB_1$-iLTD. Despite the fact that we used whole neuron lysates, our data revealed results highly consistent with other studies that isolated presynaptic mRNAs (*Bigler et al., 2017*; *Hafner et al., 2019*; *Ostroff et al., 2019*; *Shigeoka et al., 2016*), specifically, an enrichment of mRNAs encoding for initiation factors and ribosomal proteins.

We found that CB$_1$ activation significantly upregulated proteins involved in protein synthesis and processing pathways (*Figure 2*). This increase in initiation factors and ribosomal proteins suggests that plasticity likely triggers an enhanced translational capacity which is consistent with our previous findings using Fluorescent Noncanonical Amino acid Tagging (FUNCAT) (*Younts et al., 2016*). mRNAs for Arp2/3, WAVE1, and β-actin have been detected in axonal preparations and their synthesis may be important for axon formation (*Donnelly et al., 2013*; *Spillane et al., 2012*; *Wong et al., 2017*). However, in CB$_1$-iLTD, protein synthesis seems to mediate the change in presynaptic structure via the UPS, rather than direct synthesis of actin regulators. What then is being synthesized to mediate the change in structure and function? Our finding of downregulation of many presynaptic proteins and upregulation of components of the UPS suggested to us that perhaps this upregulation represents an activity-dependent synthesis of regulatory elements in the ubiquitination and proteasomal degradation pathway.

Although such a mechanism requires greater coordination, it would also reduce energy expenditure over time, that is if a presynaptic terminal will not be releasing neurotransmitter for an extended period of time (hours to days) it makes sense to degrade and recycle the release machinery, to reduce energy production, and to shrink the terminal to make space for new growth. Protein synthesis is likely necessary for the coordination and engagement of these structural, metabolic, and degradative processes. We measured changes in the protein landscape that occur fairly rapidly after CB$_1$-iLTD induction (25 min) given that CB$_1$-iLTD was dependent on protein synthesis during this time window (*Younts et al., 2016*). It is likely that additional 'plasticity-related' proteins are synthesized or degraded in the hours that follow iLTD induction.

## Protein degradation and presynaptic function

Proteomic analysis revealed a population of downregulated proteins involved in presynaptic function and structure, as well as energy metabolism (*Figures 1* and *4*). In contrast, components of the protein degradation pathway, including proteasomal subunits, E2 ubiquitin ligases, and degradative enzymes, were upregulated perhaps reflecting on-demand synthesis which could regulate fast, local presynaptic protein degradation. While activity-dependent local presynaptic synthesis of UPS machinery has never been demonstrated, there is strong evidence that presynaptic function can be regulated by expression of specific E3 ubiquitin ligases and degradation of select presynaptic proteins. For example, SCRAPPER an E3 ligase has been shown to target presynaptic proteins like RIM1, synaptophysin, and Munc18 (*Yao et al., 2007*), amongst other proteins with high similarity to those identified in our SILAC dataset. Moreover, SCRAPPER KO has been shown to impair neurotransmitter release, short-term plasticity and presynaptic LTP (*Koga et al., 2017*; *Yao et al., 2007*). The tight coupling of translation and degradation in the context of synaptic plasticity was described previously (*Klein et al., 2015*) and is presumed to occur widely in the brain as a means of maintaining proteostasis over the course of plastic changes (*Biever et al., 2019*; *Dong et al., 2008*; *Hanus and Schuman, 2013*; *Wang et al., 2017*). This rapid activity-dependent degradation could be mediated by the recently discovered neuron-specific proteasome complex (NMP) (*Ramachandran et al., 2018*), although this complex is believed to target non-ubiquitinated substrates. Protein degradation has also been shown to regulate presynaptic silencing, specifically by degradation of presynaptic proteins such as RIM1 and Munc13 (*Jiang et al., 2010*) and CB$_1$-mediated suppression of transmission at excitatory synapses via degradation of Munc18 (*Schmitz et al., 2016*; *Wang et al., 2018b*). Homeostatic plasticity in response to neuron silencing has also been associated with degradation of select presynaptic proteins (*Lazarevic et al., 2011*). However, our data strongly support the idea that protein degradation by the UPS is not directly required for CB$_1$-iLTD, but likely does occur quickly after CB$_1$ activation, as indicated by rapid loss of presynaptic proteins measured with both SILAC and western blot (*Figure 4*). This rapid reduction in protein levels presumably occurs as a consequence of enhanced UPS activity (*Figure 5*), but we cannot exclude that a different mechanism could be involved including reduced synthesis (*Dörrbaum et al., 2020*) or other protein degradation pathways, such as autophagy (*Liang and Sigrist, 2018*).

We demonstrated that ubiquitination is required for CB$_1$-iLTD, likely by controlling the trafficking, interactions, or the activity of its substrates, upstream of degradation (*Hamilton and Zito, 2013*). A previous study showed that inhibition of protein ubiquitination and degradation increased miniature EPSCs/IPSCs in cultured neurons, suggesting an important role for these processes in maintaining normal neurotransmitter release (*Rinetti and Schweizer, 2010*). However, in our hands, proteasomal

inhibitor MG-132 had no significant effects on basal synaptic transmission and neither did E1 ubiquitin ligase inhibitors, ziram and PYR-41. To our knowledge, our study is the first to describe a mechanism of long-term presynaptic structural and functional plasticity that relies on ubiquitination. The regulation of presynaptic ubiquitination is likely achieved through the targeted expression of different E2 and E3 ubiquitin ligases (*Hallengren et al., 2013*; *Koga et al., 2017*; *Yao et al., 2007*) or via presynaptic cytomatrix proteins themselves (*Chen et al., 2003*; *Ivanova et al., 2016*; *Waites et al., 2013*). This raises the possibility that presynaptic structural dynamics and UPS activity are tightly linked.

### Potential relevance in the normal and diseased brain

$CB_1$ activation via eCBs, as well as exogenous cannabinoids like $\Delta^9$- tetrahydrocannabinol (THC), the primary psychoactive ingredient in marijuana, can influence cognition, goal-directed behaviors, sensory processing and other critical brain functions (*Araque et al., 2017*; *Augustin and Lovinger, 2018*; *Häring et al., 2012*; *Heifets and Castillo, 2009*; *Hoffman and Lupica, 2013*; *Zlebnik and Cheer, 2016*). Cannabinoid signaling has also been implicated in several brain disorders (*Zou and Kumar, 2018*). Autism is broadly associated with changes in synaptic protein levels, but also disruption of $CB_1$-LTD (*Busquets-Garcia et al., 2014*; *Chakrabarti et al., 2015*). In particular, in a mouse model of Fragile X Syndrome (FXS), the most common monogenic cause of autism, where RNA-binding protein, Fragile X mental retardation protein (FMRP) is deleted (*Bagni and Zukin, 2019*), eCB-mediated plasticity in the hippocampus, striatum, prefrontal cortex is impaired (*Jung et al., 2012*; *Maccarrone et al., 2010*; *Martin et al., 2017*; *Wang et al., 2018b*; *Zhang and Alger, 2010*). Although changes in the eCB mobilization in FXS may explain some of the impairment (*Jung et al., 2012*; *Maccarrone et al., 2010*; *Zhang and Alger, 2010*), the role of FMRP in the regulation of local presynaptic protein synthesis may also play a role (but see *Jung et al., 2012*), although this remains to be tested (*Busquets-Garcia et al., 2013*). Many neurodegenerative diseases are characterized by imbalanced proteostasis, including Alzheimer's disease (AD) and Parkinson's disease (PD), resulting in pathological accumulations of misfolded proteins (*Klaips et al., 2018*). WIN treatment in animal models of AD and PD has been shown to be neuroprotective and to alleviate cognitive and motor symptoms (*Basavarajappa et al., 2017*), potentially through the ability of the $CB_1$ receptor to regulate synaptic proteostasis.

## Materials and methods

**Key resources table**

| Reagent type (species) or resource | Designation | Source or reference | Identifiers | Additional information |
|---|---|---|---|---|
| Biological sample wildtype, Sprague Dawley, *Rattus norvegicus*, male and female | Primary hippocampal neuron cultures | Charles River | | Isolated from DIV 20–21 pups |
| Biological sample wildtype, Sprague Dawley, *Rattus norvegicus*, male and female | Acute hippocampal slices | Charles River | | Isolated DOB 18–25 rats |
| Antibody | CB1 (rabbit polyclonal) | ImmunoGenes | Cat# CB1, RRID:AB_2813823 | 1:1000 |
| Antibody | Synapsin 1 (mouse monoclonal) | Synaptic Systems | Cat# 106 011C2, RRID:AB_10805139 | 1:1000 |
| Antibody | vGAT (mouse monoclonal) | Synaptic Systems | Cat# 131 011C3, RRID:AB_887868 | 1:500 |
| Antibody | Bassoon (mouse monoclonal) | Enzo Life Sciences | Cat# ADI-VAM-PS003, RRID:AB_10618753 | 1:1000 |
| Antibody | Paravalbumin | Sigma-Aldrich | Cat# P3171, RRID:AB_2313804 | 1:1000 |

*Continued on next page*

*Continued*

| Reagent type (species) or resource | Designation | Source or reference | Identifiers | Additional information |
|---|---|---|---|---|
| Antibody | α-Synuclein | BD Biosciences | Cat# 610787, RRID:AB_398108 | 1:1000 |
| Antibody | Munc-18–1 | Synaptic Systems | Cat# 116 002, RRID:AB_887736 | 1:1000 |
| Antibody | Arp2/3 | Novus | Cat# NBP1-88852, RRID:AB_11040464 | 1:1000 |
| Antibody | Ubiquitin K48 | Millipore | Cat# 05–1307, RRID:AB_1587578 | 1:1000 |
| Software | Igor Pro | IGOR Pro | RRID:SCR_000325 | |

## Immunohistochemistry and microscopy

Acute rat hippocampal slices were made as described below for electrophysiological recordings and allowed to recover for at least 1 hr after slicing. Slices were incubated in beakers containing ACSF and drug treatments described in Results and underwent constant oxygenation. Slices were fixed immediately after treatments in 4% PFA in PBS overnight at RT. Slices were washed twice in PBS then incubated in blocking buffer (4% BSA in PBS + 0.1% Tx-100) for 1 hr at RT. Primary antibodies (CB$_1$, 1:1000, Immunogenes (Budapest, Hungary)); Synapsin-1 1:1000 Synaptic Systems (Goettingen, Germany); Bassoon, 1:1000, Enzo Life Sciences (Farmingdale, NY); Paravalbumin, 1:1000, Sigma Aldrich; VGAT, 1:500, Synaptic Systems were diluted directly into the blocking buffer and floating slices were incubated overnight at 4C. After four washes with PBS, slices were incubated in secondary antibodies (Invitrogen) diluted in blocking buffer overnight at 4°C. Slices were washed 5X with PBS, then mounted. Images were acquired on a Zeiss LSM 880 with Airyscan using a Plan-Apochromat 63x/1.4 Oil DIC M27 and 1.8X zoom. Images were Airyscan processed prior to analysis. Pixel width and height was 0.049 μm and voxel depth was 0.187 μm. Imaris 9.2 software was used to reconstruct boutons in 3D using the Surface function. Threshold, laser power, and gain were kept constant for each experiment. CB$_1$ boutons were screened after 3D reconstruction to ensure correct identification. Only boutons that fell between 0.05–5 μm$^3$, did not touch the image border, and had a sphericity value above 0.3 were considered. For Bassoon (*Figure 1F*) and VGAT (*Figure 1—figure supplement 2D*), FIJI was used to remove all Bassoon signal that did not overlap with CB$_1$ labeling by creating a dilated binary mask of CB$_1$ labeling then using the Image Calculator 'AND' function to create a mask for the non-CB$_1$ channel, then this mask was used to isolate signal in the non-CB$_1$ channel, then Imaris was used to measure the volume of the non-CB$_1$ channel. FIJI was used to analyze synapsin puncta (*Figure 4D*) inside CB$_1$ boutons by creating a dilated binary mask of CB$_1$ labeling then using the Image Calculator 'AND' function to create a mask for the non-CB$_1$ channel, followed by the 'Analyze Particles' function to determine the intensity or percent overlap of the two channels. All imaging and analysis were performed blind to treatment group.

## SILAC

Primary hippocampal neurons were prepared from E18-19 rat brains and grown on poly-D-lysine coated 15 cm plates at a density of 3.5 million cells/ 15 cm plate in DMEM media without l-arginine or l-lysine (Cambridge Isotope Laboratories, Tewksbury, MA, USA, Cat# DMEM-500), with pen/strep and B-27 supplement (Invitrogen, Carlsbad, CA, USA). 84 mg/L of l-arginine 13 $_{C6}$ (Sigma Aldrich, St. Louis, MO) and 146mg/L of l-lysine D$_4$ (Thermofisher, Waltham, MA) was supplemented for 'medium' labeled media and 84 mg/L of l-arginine 13 $_{C6}$615$_{N4}$ and 146 mg/L of l-lysine 13 $_{C6}$615$_{N4}$ was added to 'heavy' labeled media. Neurons were grown in this media for 15 days, which results in >90% incorporation of labeled amino acids into the cellular proteomes (*Zhang et al., 2011*; *Zhang et al., 2012*). For treatment, neuron cultures DIV 16 received 15 ml fresh media containing either vehicle or WIN (5 μM) for 25 min. Neurons were washed 3X with ice-cold PBS without Mg$^{2+}$ or Ca$^{2+}$ (0.01 M, pH = 7.4). After three washes, cells were harvested and lysed in SDS lysis buffer containing: (50 mM Tris, 2% SDS, 2 mM EDTA) for 30 min at RT. Lysates were then sonicated briefly, allowed to incubate for another 30 min at RT, and centrifuged for 5 min at 15,000 g to remove insoluble debris. 10 μg of lysate from 'medium' cells treated with vehicle were mixed with 10 μg 'heavy'

WIN-treated cells (Forward sample). Separately, 10 µg of lysate from 'medium' cells treated with WIN were mixed with 10 µg 'heavy' vehicle-treated cells (Reverse sample). These mixtures (20 µg total protein) were loaded onto a 10% Bis-Tris gel and subjected to SDS-PAGE. The gel was stained with Coomasie for 1 hr and protein lanes were cut up into 12 equal sized portion in order to improve protein coverage (*Jordan et al., 2004*). Mass spectrometry was performed in collaboration with the Einstein Proteomic Facility using the Orbitrap Velos. In Mascot the Quantitation Method (SILAC K+4 K+8 R+6 R+10) was used with each SILAC modification in exclusive mode. Those listed as variable were for determining incomplete protein labeling: 2H(4) (K); 13C(6) (R); 13C(6)15N(2) (K); and 13C(6) 15N(4) (R). The raw data files were first processed using precursor ion quantitation of the Quantitation toolbox of Mascot Distiller (Matrix Science Ltd; version 2.7). Mascot was then used to search the rat databases (SwissProt and NCBInr along with a decoy database to obtain FDRs) using the following parameters: trypsin; product ion mass tolerance of 0.40 Da; precursor ion tolerance of 50 PPM; carbamidomethyl Cys - fixed modification and variable modifications of: deamidated Asn and Gln; label:2H(4) of Lys; label:13C(6) of Arg; label:13C(6)15N(2) of Lys; label:13C(6)15N(4) of Arg and oxidation of Met. The result files obtained from Distiller and the Mascot searches were then uploaded to Scaffold Q+S (Proteome Software Inc; version 4.9) using between-subjects, $\log_2$ ratio-based analysis of unique peptides against a reference to obtain the protein's mean quantitative values and t-test p-values with Benjamini-Hochberg correction. Peptide identifications were accepted if they could be established at greater than 95.0% probability by the Scaffold Local FDR algorithm. Protein identifications were accepted if they could be established at greater than 99.0% probability and contained at least two identified peptides. Protein probabilities were assigned by the Protein Prophet algorithm (*Nesvizhskii et al., 2003*). Proteins that contained similar peptides and could not be differentiated based on MS/MS analysis alone were grouped to satisfy the principles of parsimony. Proteins sharing significant peptide evidence were grouped into clusters. The mass spectrometry proteomics data have been deposited to the ProteomeX change with identifier Consortium via the PRIDE [1] partner repository with the dataset identifier PXD020008 and 10.6019/PXD020008.

## Gene ontology analysis

SILAC results were ranked according based on fold change and submitted to a GSEA Preranked analysis in GSEA (v. 4.0.2) with 1000 permutations. Terms smaller than 15 genes or bigger than 500 were discarded as previously reported (*Merico et al., 2010*). The enrichment map was generated in Cytoscape (3.7.1) (*Kucera et al., 2016*) using Enrichment map plugin (3.2.0) (*Merico et al., 2010*) using the following thresholds: p value < 0.05, FDR < 0.001. The overlap coefficient was set to 0.5. For confirmation, we also performed Gene Ontology analysis using two other tools. First, filtered lists ($|\log_2$ fold change$| > 0.5$) were analyzed through the use of IPA (QIAGEN Inc, Hilden, Germany, https://www.qiagenbioinformatics.com/products/ingenuitypathway-analysis). Then, we performed ontology enrichment using a recently published expert-curated knowledge database for synapses (*Koopmans et al., 2019*). Terms were selected with a FDR < 0.01. The parental term 'Synapse' was discarded as not being informative (e.g. to general).

## Electrophysiology slice preparation and recording

Experimental procedures adhered to NIH and Albert Einstein College of Medicine Institutional Animal Care and Use Committee guidelines. Acute transverse slices were prepared from young adult male and female Sprague Dawley rats (P18-27). The cutting solution contained (in mM): 215 sucrose, 20 glucose, 26 $NaHCO_3$, 4 $MgCl_2$, 4 $MgSO_4$, 1.6 $NaH_2PO_4$, 2.5 KCl, and 1 $CaCl_2$. The artificial cerebral spinal fluid (ACSF) recording solution contained (in mM): 124 NaCl, 26 $NaHCO_3$, 10 glucose, 2.5 KCl, 1 $NaH_2PO_4$, 2.5 $CaCl_2$, and 1.3 $MgSO_4$. After ice-cold cutting, slices recovered at RT (in 50% cutting solution, 50% ACSF) for <30 min and then at room temperature (RT) for 1 hr in ACSF. All solutions were bubbled with 95% $O_2$ and 5% $CO_2$ for at least 30 min. Although the form of long-term inhibitory synaptic plasticity studied here (i.e. iLTD) is present under physiological recording conditions at 37°C (*Younts et al., 2013*), inhibitory synaptic transmission is less stable at this temperature, and therefore we conducted our experiments at 25.5 ± 0.1°C.

For extracellular field recordings, a single borosilicate glass stimulating pipette filled with ACSF and a glass recording pipette filled with 1M NaCl were placed approximately 100 µm apart in *stratum pyramidale*. To elicit synaptic responses, paired, monopolar square-wave voltage or current

pulses (100–200 µs pulse width) were delivered through a stimulus isolator (Isoflex, AMPI) connected to a broken tip (~10–20 µm) stimulating patch-type micropipette filled with ACSF. Typically, stimulating pipettes were placed in CA1 stratum pyramidale (150–300 µm from the putative apical dendrite of the recorded pyramidal cell, 150–200 µm slice depth). Stimulus intensity was adjusted to give comparable magnitude synaptic responses across experiments less than ~0.6 mV. Inhibitory synaptic transmission was monitored in the continuous presence of the NMDA receptor antagonist d-(-)−2-amino-5-phosphonopentanoic acid (d-APV; 25 µM), the AMPA/kainate receptor antagonist 2,3-dihydroxy-6-nitro-7-sulfonyl-benzo[f]quinoxaline (NBQX; 5 µM), and the µ-opioid receptor agonist, [D-Ala$^2$, N-MePhe$^4$, Gly-ol]-enkephalin (DAMGO, 50 nM). To elicit chemical-iLTD (*Heifets et al., 2008*) the $CB_1$ agonist WIN 55,212–2 (WIN; 5 µM) was bath applied for 25 min, and 5 stimuli at 10 Hz were delivered at 0.1 Hz during the last 10 min of WIN. WIN was chased with the $CB_1$ inverse agonist/antagonist SR 141716 (5 µM) or AM251 (5 µM) to halt $CB_1$ signaling. Baseline and post-induction synaptic responses were monitored at 0.05 Hz during iLTD. Stimulation and acquisition were controlled with IgorPro 7 (Wavemetrics). Shaded boxes in figures correspond to when plasticity was analyzed with respect to baseline and when representative traces were collected and averaged. Summary data (i.e. time-course plots and bar graphs) are presented as mean ± standard error of mean (S.E.M.). PPR was defined as the ratio of the amplitude of the second EPSC (baseline taken 1–2 ms before the stimulus artifact) to the amplitude of the first EPSC. The magnitude of LTD was determined by comparing 20 min baseline responses with responses 80–100 min post-LTD induction.

## Western blotting

Protein concentration was determined using the Lowry method with bovine serum albumin as a standard (*Lowry et al., 1951*). Primary hippocampal neurons or hippocampal slices were solubilized on ice with RIPA buffer (1% Triton X-100, 1% sodiumdeoxycholate, 0.1% SDS, 0.15MNaCl, 0.01Msodium phosphate, pH7.2) followed by sonication. Immunoblotting was performed after transferring SDS-PAGE gels to nitrocellulose membrane and blocking with 5% low-fat milk for 1 hr at room temperature. The proteins of interest were visualized after incubation with primary antibodies (α-synuclein 1:1000 BD Biosciences (San Jose, CA) #610787; Synapsin-1 1:1000 Synaptic System (Goettingen, Germany) #106001; Munc18–1 1:1000 Synaptic System #116 002; Arp2/3 1:1000 Novus Biologicals (Centennial, CO) # NBP188852; Ubiquitin K48 1:1000 EMD Millipore (Burlington, MA) #05–1307) by chemiluminescence using peroxidase-conjugated secondary antibodies in LAS-3000 Imaging System (Fujifilm, Tokyo, Japan). Densitometric quantification of the immunoblotted membranes was performed using ImageJ (NIH). All protein quantifications were done upon normalization of protein levels to Ponceau staining. Ponceau normalization was chosen over comparison to actin as our work and others showed that $CB_1$-iLTD induces modification of actin cytoskeleton.

## Ubiquitination sites analysis

Ubiquitination sites were identified using Ubisite, a publicly available resource for ubiquitination site prediction (*Akimov et al., 2018*). To minimize false positive rate, confidence level was set on high. When available, functional domains were annotated using the uniport.org database (*UniProt Consortium, 2019*).

## Data analysis, statistics and graphing

Analysis and statistics were carried out in OriginPro 2015 (OriginLab) and Graphpad Prism 7.02. Significance (p<0.05) was assessed with one-way ANOVA (means comparison with *post hoc* Tukey test), Student's paired and unpaired t-tests, Wilcoxon matched-pairs signed rank test, Mann Whitney U test, or Pearson's correlation coefficient, as indicated. All electrophysiology experiments were performed in an interleaved fashion –that is control experiments were performed every test experiment. Unless stated otherwise, 'n' represents number of field recordings in slices. All experiments include at three animals. Plotting of SynGO results was made using matplotlib (3.0.3)(*Hunter, 2007*) in Python (3.7. 3)(*Oliphant, 2007*) environment. *Figure 4D* used Garder-Altman estimation plots to represent effect size. Statistics and graphing were performed using estimationstats.com (*Ho et al., 2019*). Supplementary figures include Superplots to represent individual bouton values color-coded by slice that were created using Python (3.7.3) (*Lord et al., 2020*) and informed by plot design from

the Superplots app (https://huygens.science.uva.nl/SuperPlotsOfData/). Slice numbers were balanced across conditions in some Superplots (in *Figure 1—figure supplement 1A*, *Figure 3—figure supplement 1A*, and *Figure 5—figure supplement 1E*) to make statistical comparison across similar sized groups (*Lord et al., 2020*), and to make it easier for the reader to compare the distribution of individual bouton values across conditions. To balance the conditions in an unbiased manner, if a condition had an unequal number of slices, the slice that had the most different number of 'n' was removed. Importantly, balancing the conditions did not change the interpretation of the data as plotted in the main figures and all key observations were reproduced.

## Reagents

Stock reagents were prepared according to the manufacturer's recommendation in water, DMSO (<0.01% final volume during experiments), or phosphate buffered saline (PBS), stored at −20°C, and diluted into ACSF or intracellular recording solutions as needed. CNQX, D-APV, SR 141716, and WIN 55,212–2 were acquired from the NIMH Chemical Synthesis and Drug Supply Program; salts for making cutting, ACSF, ziram, and intracellular recording solutions from Sigma Aldrich (St. Louis, MO); AM251, NSC-23766, MG-132, DAMGO, cycloheximide from Tocris Bioscience (Bristol UK); jasplakinolide, anisomycin, PYR41 from Cayman Chemical (Ann Arbor, MI); CK-666 from EMD Millipore. Reagents were either acutely bath applied, diluted into the intracellular recording solution, or preincubated with slices/cultures, as indicated in Results.

## Acknowledgements

We thank Dr. Thomas Younts, Dr. Matthew Klein, and Dr. Ana Maria Cuervo for helpful discussions. We thank Dr. Kostantin Dobrenis, Kevin Fisher, and Vladimir Mudragel of the Einstein Neural Cell Engineering and Imaging Core (supported by The Rose F Kennedy Intellectual Disabilities Research Center) for their advice and assistance with Airyscan confocal microscopy acquisition and analysis. We thank Edward Nieves of the Einstein Proteomics Facility for assistance performing and analyzing proteomic data. This research was supported by the National Institutes of Health: F31MH114431 to HRM, R01-MH081935, R01-DA17392, R01-NS113600 and a pilot grant through NICHD U54 HD090260 to PEC, a shared instrument grant (1S10OD25295) to KD, R01-AG039521 to BAJ. MB was supported by the Rainwater Charitable Foundation. HM and MB also received Junior Investigator Neuroscience Research Award (JINRA) from Dominick P Purpura Department of Neuroscience at Albert Einstein College of Medicine for this project.

## Additional information

### Funding

| Funder | Grant reference number | Author |
| --- | --- | --- |
| National Institute of Mental Health | F31MH114431 | Hannah R Monday |
| National Institute of Mental Health | R01-MH081935 | Pablo E Castillo |
| National Institute on Drug Abuse | R01-DA17392 | Pablo E Castillo |
| National Institute of Neurological Disorders and Stroke | R01-NS113600 | Pablo E Castillo |
| National Institute on Aging | R01-AG039521 | Bryen A Jordan |
| Rainwater Charitable Foundation | | Mathieu Bourdenx |

The funders had no role in study design, data collection and interpretation, or the decision to submit the work for publication.

## Author contributions
Hannah R Monday, Conceptualization, Resources, Data curation, Software, Formal analysis, Supervision, Funding acquisition, Validation, Investigation, Methodology, Writing - original draft, Writing - review and editing; Mathieu Bourdenx, Conceptualization, Resources, Data curation, Software, Formal analysis, Funding acquisition, Visualization, Writing - review and editing; Bryen A Jordan, Supervision, Methodology, Writing - review and editing; Pablo E Castillo, Conceptualization, Resources, Supervision, Funding acquisition, Writing - original draft, Writing - review and editing

## Author ORCIDs
Hannah R Monday ⓘ https://orcid.org/0000-0003-3969-492X
Pablo E Castillo ⓘ https://orcid.org/0000-0002-9834-1801

## Ethics
Animal experimentation: Experimental procedures adhered to NIH and Albert Einstein College of Medicine Institutional Animal Care and Use Committee guidelines as approved by protocol #00001047.

## Decision letter and Author response
Decision letter https://doi.org/10.7554/eLife.54812.sa1
Author response https://doi.org/10.7554/eLife.54812.sa2

# Additional files

## Supplementary files
- Supplementary file 1. Full protein list and Ingenuity Pathway Analysis.
- Supplementary file 2. Raw data for GSEA and enrichment map analysis.
- Transparent reporting form

## Data availability
All data generated in this study are included in the manuscript and supporting files. Source data files are provided for Figure 2. The mass spectrometry proteomics data have been deposited to the ProteomeX change with identifier Consortium via the PRIDE [1] partner repository with the dataset identifier PXD020008 and https://doi.org/10.6019/PXD020008.

The following dataset was generated:

| Author(s) | Year | Dataset title | Dataset URL | Database and Identifier |
|---|---|---|---|---|
| Monday HR, Bourdenx M, Jordan BA, Castillo PE | 2020 | CB1 receptor-mediated inhibitory LTD triggers presynaptic remodeling via protein synthesis and ubiquitination | https://www.ebi.ac.uk/pride/archive/projects/PXD020008 | PRIDE, PXD020008 |

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
