## [Decision Letter]

**Acceptance summary:**

This study reveals structural plasticity of presynaptic terminal bouton sizes and underlying molecular mechanisms for endocannabinoid induced long-term depression of inhibitory synapses (iLTD) in the hippocampus. The results establish that the well-described postsynaptic plasticity-related spine growth/shrinkage can also exist in presynaptic compartments in presynaptic forms of plasticity and provide insights into this relatively less understood aspect of synaptic plasticity.

**Decision letter after peer review:**

Thank you for submitting your article "CB_1_ receptor-mediated inhibitory LTD triggers presynaptic remodeling via protein synthesis and ubiquitination" for consideration by *eLife*. Your article has been reviewed by three peer reviewers, and the evaluation has been overseen by a Reviewing Editor and Gary Westbrook as the Senior Editor. The following individual involved in review of your submission has agreed to reveal their identity: Erin M Schuman (Reviewer #1).

The reviewers have discussed the reviews with one another and the Reviewing Editor has drafted this decision to help you prepare a revised submission.

The reviewers were positive about the manuscript but had a robust discussion of points that they though required additional input. The summary includes the main points that arose in the discussion. The original reviews follow. Please address both the summary and the reviews in your revised manuscript.

Summary of reviewer discussion:

1) Immunostaining for the CB_1_ receptor was used as a proxy for terminal size. This could be OK, but the bouton volume data should be checked critically. For example, the shrinkage of the terminal as measured by Bassoon is smaller than that measured by CB_1_. The authors use an agonist of CB_1_ throughout the paper – this will certainly change the configuration of the receptor (dimerization, clustering, internalization) that could artificially inflate the shrinkage they claim. This point should be discussed.

2) The MS data needs the tolerance value and stats. The link between the requirement for ubiquitination and protein synthesis is not as strong as it could be. The authors could add experimental data or should at least include a working model in their discussion addressing such questions as: Are both required? Do they operate in parallel or is new synthesis needed for ubiquitination?

3) Drug alone controls are missing for several conditions, particularly CHX and JSK.

4) The conclusions RE Rac and Arp need to be modified because the cellular site of action is not established in this study. The manipulations are pharmacological, and although the narrative and proteomics wants us to think about a pre-terminal action, one could imagine postsynaptic effects as these proteins are well-known for postsynaptic roles. The authors need to modify their conclusions or use tools that exist to cell specifically manipulate these proteins and establish the site of action.

Reviewer #1:

This manuscript explores a functional role for presynaptic protein synthesis in modulating structural plasticity of inhibitory presynaptic boutons during LTD. The authors provide data for protein synthesis dependent shrinkage of CB_1_+ boutons during endocannabinoid mediated LTD. Using proteomics, they identify 33 upregulated and 27 downregulated proteins during LTD. They narrow in and explore a role for actin remodeling during LTD, focusing on the role of Rac1 and Arp2/3, specifically the downregulation of both, for the manifestation of CB_1_+ boutons shrinkage. Lastly, they argue that while proteasomal degradation is induced during LTD, only the ubiquitination itself is important for the structural and functional manifestation of LTD.

The data presented on presynaptic structural plasticity are exciting but there are missing key control experiments and validations – as indicated below.

Figure 1:

While the detailed description of the analysis of the boutons was appreciated, the data presented in 1C are not convincing in their present state. The authors provide no controls to ensure that it's not endocytosis/remodeling of the CB_1_ receptors themselves, which would also result in a decreased CB_1_ volume. Additional inhibitory presynaptic markers (such as VGAT) should also be assessed and scored. Given the strict cutoffs the authors use in the analysis (ie sphericity value within the Imaris analysis), it could be that they're simply detecting a subset of smaller CB_1_ puncta resulting from local changes in the receptor pool; while the boutons themselves remain largely unchanged. The use of this CB_1_ volumetric analysis through the paper is problematic, as in almost all cases the authors do not test the effects of the pharmacological treatment alone (CHX, JSK, CK-666, NSC and ziram). Most if not all of the conclusions could be substantially undercut if these treatments in control conditions show the same change as when paired with WIN treatment. The study of structural plasticity in the postsynapse has largely relied on live cell imaging, so the compartment can be analyzed before, during and post treatment. While challenging, the authors should consider a similar approach, even if only to demonstrate their CB_1_ volumetric approach is valid. Using either an inhibitory neuron restricted GFP expression or expression of a tagged inhibitory presynaptic protein to allow live imaging of the bouton volume before and after WIN application, the authors could the perform fixation and staining for CB_1_ receptors and analyze how this compartment changed during the course of the treatments. This would bring their approach more in line with the standard in the field and allow for much more current and compelling analysis for structural plasticity to be performed.

Specific comments:

– Examples (similar as in 1A) should be provided of CB_1_ boutons with WIN/WIN+CHX treatment.

– CHX treatment alone should be assessed for C.

– CHX and WIN+CHX should be assessed for D.

– Did the authors look at what happens to the corresponding inhibitory postsynaptic compartments? Presumably these also may decrease in size as well.

– For Figure 1—figure supplement 1C, the indicated mean for the WIN group for the graph provided should be checked. Given the large increase in the low volume population seen in the WIN group vs control group; and the similar size of the groups (540 vs 602) it's hard to understand how both means can be right around 1.

Figure 2:

The experimental design (multiplexing of fully medium-heavy and heavy labeled cell cultures) is nice, but there are several potential issues regarding data processing and interpretation.

Protein identification and quantification:

– The accepted parent mass tolerance of 50 PPM is very high and might lead to false identifications. When acquiring the data with an MS1 resolution of 60k on a well-calibrated Orbitrap LTQ instrument, a MS1 mass tolerance of ~5 PPM should be used.

– Why are the SILAC labels (Lys4/Arg6 and Lys8/Arg10) specified as variable modifications instead of “SILAC pairs”? Please describe in more detail how the medium-heavy and heavy counterparts were quantified and how the fold-changes were calculated. Were only peptides species considered that were quantified in the medium heavy and heavy form? Why not using software that provide well-established workflows for SILAC quantification (such as the freeware MaxQuant)?

– From Supplementary file 1, we understand that each ~700 proteins were identified in the forward and the reverse samples. The overlap between the two samples is 391 proteins (main text), indicating that the reproducibility is fairly low between the measurements.

Statistics:

– Please perform proper statistics to identify differentially regulated proteins (e.g. t-test and multiple testing correction). A simple fold-change cut-off (as described in Figure 2C) is not state-of-the-art.

Specific comments:

– The supplementary data should be improved to enable the reader to understand the proteomics data and inspect all quantified proteins. Please include one table that presents the quantitative results for all proteins in all biological replicates (forward and reverse), so that the reader can get a sense for the biological variability. Please include a column with gene names. In Supplementary file 1, why does the table contain proteins from *rattus norvegicus*, *mus musculus* and human?

– All mass spec raw data and database search results should be uploaded to the PRIDE repository prior to publication and the PRIDE identifier should be included in the Materials and methods section.

Data interpretation:

– Protein up-regulation is often interpreted as increased protein synthesis and protein down-regulation is often interpreted as increased degradation in the manuscript. However, based on the SILAC-MS data one cannot draw these conclusions. Protein up-regulation could result from more synthesis or less degradation and protein down-regulation could result from less synthesis or more degradation. Based on the presented data there is no evidence for one or the other.

– Based on the proteomics data one cannot distinguish proteins from different subcellular compartment or different cell types. The following statement from the Discussion sections is hence misleading: "[…] why ribosomal proteins would be synthesized locally if ribosomes are exclusively assembled in the nucleolus is unclear."

Figure 3:

The authors refer to JSK as an "actin stabilizing drug", whereas in fact it's a drug that promotes actin polymerization. This fact complicates the data presented in this figure as essential controls (such as JSK treatment alone for 3A) are missing. If JSK treatment alone increases the bouton size, is this really demonstrating that the structural remodeling requires actin depolymerization?

Specific comments:

– 3A, the examples provided with the black background are harder to examine. The authors should use the white background used in Figure 1 and Figure 1—figure supplement 1 examples.

– 3B, what explanation do the authors have for why the +JSK/WIN treated curve stabilizes around 80%?

– Additionally, the authors should test the effect of the Arp2/3 inhibitor and Rac inhibitor with and without WIN treatment on the bouton volume.

Figure 5:

Specific comments

– 5D, examples would be better with white backgrounds.

– Analysis for the effect of ziram alone needs to be done.

Reviewer #2:

In this study by Monday and colleagues, the investigators reveal structural plasticity of presynaptic terminal bouton sizes and underlying molecular mechanisms for iLTD, a form of endocannabinoid induced long-term depression of inhibitory synapses. Their discovery establishes that the well-described post-synaptic phenomenon of spine growth/shrinkage with plasticity can also exist in presynaptic compartments in presynaptic forms of plasticity. This finding is demonstrated by volumetric bouton measurements of CB_1_+ terminals in hippocampal slices. The remainder of the study involves a series of elegantly and rigorously designed experiments to provide insights to the molecular mechanisms for this structural plasticity. Insights begin with a proteomic analysis performed on cultured hippocampal neurons, and then use pharmacology and Western analysis to test a main hypothesis that changes in the abundance and activity of actin regulators underlie the structural plasticity. The study continues to reveal that despite their hypothesis that proteasomal degradation could drive those shifts, experiments instead support the idea that ubiquitination, in the absence of degradation (assessed by MG-132) appears to underlie the functional differences required for iLTD.

Overall, this study provides an important extension to the Castillo lab's recent work demonstrating the requirement for local protein synthesis for this form of plasticity (Younts et al., 2016). Strengths of this study include the mechanistic depth, general rigor in approaches, and novel insights to a relatively less understood aspect of synaptic plasticity.

I have two major concerns that should be addressed.

1) Boutons simply should not be considered individual n's for statistics. The authors are welcome to argue for what the independent biological unit is. Please also clarify, for a given condition (such as Veh or WIN), were slices from the same animal used for multiple conditions, or were the slices from one animal used for one condition at a time? If the latter, then an animal is the biological unit, with the possibility of using nested data statistical approaches. Alternatively, the mean value/slice could be considered. Issues such as animal health, temperature fluctuations, buffer osmolarity et cetera could otherwise exist and generate 100s of values per slice that are a little bigger or smaller due to technical variables. This is a foundational observation for the study and needs to be robust. As presented, it looks small and only visible when n's are in the 1000s. Likewise, although it was good to consider involvement of other inhibitory synapses (like PV^+^), a figure showing no significant difference that is potentially grossly underpowered is not so useful (n's of 500).

2) In general, the bioinformatics approach was rigorously and thoroughly presented. It is also nice that the authors further discuss results from other programs. One item to clarify – since the GO analyses are the major data presented: please review and clarify the extent to which the *same* protein is annotated with multiple functional properties such that it is populating (and driving) multiple terms on those lists.

Reviewer #3:

In this manuscript the authors show that CB_1_-LTD involves presynaptic structural plasticity: bouton volumes are reduced, which occurs at least partially via actin reorganization.

Mass spec data showed results that may have been expected (downregulation of synaptic proteins), but also some unexpected results were found (upregulation of protein synthesis as well as protein degradation system). Finally, the authors show that activation of the ubiquitination system is required for CB_1_-LTD, however not by promoting protein degradation by the proteasome.

The manuscript contains an interesting data set and the involvement of the ubiquitination system in CB_1_-LTD is highly intriguing. However, after reading the manuscript there are many “lose ends”, and it is still not clear what happens during CB_1_-LTD. Indeed, the authors themselves describe these events as “coordinated engagement of multiple cellular processes”, which does not clarify much. I would like to encourage the authors to try and formulate a more complete hypothesis of the cellular events that underlie CB_1_-LTD.

I have two major points:

1) It is not easy to link the proteomics findings to changes in the actin cytoskeleton in presynaptic boutons. For instance, the authors show that WIN slightly reduces Arp2c protein levels, but that inactivating Arp2/3 actually enhances CB_1_-LTD. I am wondering if the explanation given by the authors the most plausible interpretation of the data. Is it known how CB_1_-LTD is expressed (fewer vesicles? reduced release probability?) and is it possible to link this to specific changes in the actin cytoskeleton?

2) The proteomics data show that multiple processes are activated during CB_1_-LTD. The manuscript does not provide a model or hypothesis for how these processes are linked. For instance, are ubiquitination and protein synthesis independently regulated by CB_1_? What is the link between ubiquitination and presynaptic actin remodeling? Would it be possible to link CB_1_-LTD to a specific E2 or E3 ligase in the presynapse?

[Editors' note: further revisions were suggested prior to acceptance, as described below.]

Thank you for submitting your revised article " CB_1_ receptor-mediated inhibitory LTD triggers presynaptic remodeling via protein synthesis and ubiquitination" to *eLife*. Your article has been reviewed by three peer reviewers, and the evaluation has been overseen by Gary Westbrook as the Senior Editor. The following individuals involved in review of your submission have agreed to reveal their identity: Erin M Schuman (Reviewer #1).

The reviewers have discussed the reviews with one another and the Senior Editor. There are some remaining concerns raised by one of the reviewers and by me that we would like you to address as below. These are concerns about the data presentation as embodied in comment 1 of reviewer 2 in the original review. The issue concerns what constitutes "n" in the bouton counts. We suggest a compromise in which supplemental data are shown by slice for full data transparency. Overall, we suspect that artifact is not the basis of the findings, but the data presentation does not allow that assessment and therefore lowers the rigor and potential reproducibility.

Requested revisions

1) The authors in their rebuttal argue that standard power calculations are "unreasonable" because n of 145 slices would be required. At the most fundamental level, if this were true, this question and conclusion are not appropriate given the constraints of the methodology and one should focus on powerable questions. However, a statistical analysis by a statistician might indicate other analytical approaches. The authors show how close to being powered they may be with the slice means alone (not including nesting) showing significance. If the authors want to show the data as is in the main figure, the authors could address this concern by creating supplemental figures that show bouton size distributions for each slice and also figures with slice means and exact p values.

2) Bassoon puncta were to serve as an independent measure of bouton size to support the CB_1_ observations. However, the methods describe first making a binary mask of CB_1_ signal, then using the "and" function to isolate the bassoon pixels that overlap with CB_1_ pixels. As described, it is almost impossible to reach any other conclusion than reproducing the CB_1_ measurement results. Please clarify methods substantially to remove this concern.

3) In working to propose a solution for data presentation, the authors raise an important point, supplemental data should include an analysis that addresses whether the plasticity effects preferentially involve a particular size subset, smaller or large.

[Editors' note: further revisions were suggested prior to acceptance, as described below.]

Thank you for submitting your article " CB_1_ receptor-mediated inhibitory LTD triggers presynaptic remodeling via protein synthesis and ubiquitination" for consideration by *eLife*. Your article has been reviewed by one peer reviewer, and the evaluation has been overseen by Gary Westbrook as the Senior Editor. Before final acceptance, please edit the text and Materials and methods to indicate your response to the final reviewer comments as below. I would appreciate it if you indicate in the text where the revisions have been made. The final submission will not go back to the reviewers.

Reviewer #2:

1) The supplementary figures are highly appreciated and allow critical review of the nature of the experimental data and robustness of the conclusions.

2) The authors' response indicating that data was removed to balance slice numbers needs to be clarified. One should not have to remove data. If slices were removed from supplementary figure analyses, what guided those decisions? Do the removed slices remain in the main figure data? I would suggest all the data that were believed to be valid in the original submission be included in the main and supplementary figures. (Or was there a situation of grossly unequal n's in the control/win condition and the authors are trying to make statistical comparisons across similarly sized groups post hoc? In principle, a similar number of interleaved controls should exist for each additional test condition.) Please clarify.

---

## [Author Response]

Summary of reviewer discussion:1) Immunostaining for the CB_1_ receptor was used as a proxy for terminal size. This could be OK, but the bouton volume data should be checked critically. For example, the shrinkage of the terminal as measured by Bassoon is smaller than that measured by CB_1_. The authors use an agonist of CB_1_ throughout the paper – this will certainly change the configuration of the receptor (dimerization, clustering, internalization) that could artificially inflate the shrinkage they claim. This point should be discussed.

The bouton volume and active zone data has been thoroughly checked. We have addressed the reviewers concerns about using CB_1_ as a proxy for bouton volume in the following ways: 1. While CB_1_ internalization has been reported, it does not require protein synthesis, so the fact that we see a blockade of the structural change with cycloheximide would suggest we are measuring a change in volume, not internalization of CB_1_. 2. Based on previous work, it seems unlikely that our protocol would cause significant redistribution of CB_1_ (Dudok et al., 2015; Thibault et al., 2013). In particular, the very high density of CB_1_ on interneuron boutons makes it such that even a partial shift of CB_1_ to endosomes would not affect the labeling of the bouton membrane (Thibault et al., 2013). 3. We have quantified the colocalization of CB_1_ and endosomal marker, Rab7 (Feng et al., 1995), a way of estimating internalized CB_1_ (Howlett et al., 2010), which was not significantly altered by CB_1_-iLTD induction (see Author response image 1 in response to reviewer 1 below) 4. We do not anticipate that clustering or dimerization could artificially inflate the shrinkage because these processes would not necessarily pull receptors towards the bouton centerpoint, but would more likely stochastically change the distribution of CB_1_ on the surface of the membrane which, given our 3D analysis, would still provide an accurate measure of the bouton volume. Moreover, dimerization/clustering should not be sensitive to protein synthesis inhibition. 5. Finally, and as suggested by reviewer 1, we have added new data demonstrating that VGAT, an additional CB_1_-independent marker of inhibitory boutons, is also robustly reduced by CB_1_ (Figure 1—figure supplement 1E). 6. Lastly, we have now added a thorough discussion of the research that exists on CB_1_ internalization.

In addition, as requested, we have added new data in which we analyzed the Bassoon size following cycloheximide treatment (See Figure 1F). In this dataset, which now includes more “n’s”, the magnitude of the reduction in active zone size following WIN treatment is similar to that measured with CB_1_. However, it is not clear whether all components of the presynaptic bouton shrink proportionately. For example, we see a fairly large change in VGAT volume, consistent with previous reports of short-term CB_1_ activation-induced vesicle redistribution and changes in numbers of docked/primed vesicles (Garcia-Morales et al., 2015; Ramirez-Franco et al., 2014).

2) The MS data needs the tolerance value and stats. The link between the requirement for ubiquitination and protein synthesis is not as strong as it could be. The authors could add experimental data or should at least include a working model in their discussion addressing such questions as: Are both required? Do they operate in parallel or is new synthesis needed for ubiquitination?

In response, we now include the tolerance value (added in Materials and methods section) and results of statistical comparison between vehicle and treatment conditions. We have also included our proposed model for the molecular mechanisms underlying CB_1_-iLTD (Figure 5E). Our data demonstrate that CB_1_ activation triggers protein synthesis, protein degradation and structural changes. Our SILAC data clearly show an increase in several proteins that are involved in proteasomal function, suggesting that protein synthesis is upstream of protein degradation, and not in parallel.

3) Drug alone controls are missing for several conditions, particularly CHX and JSK

The drug alone controls have now been included for CHX and JSK (Figure 1—figure supplement 1A, Figure 1E, 3A) and also for ziram (Figure 5D).

4) The conclusions RE Rac and Arp need to be modified because the cellular site of action is not established in this study. The manipulations are pharmacological, and although the narrative and proteomics wants us to think about a pre-terminal action, one could imagine postsynaptic effects as these proteins are well-known for postsynaptic roles. The authors need to modify their conclusions or use tools that exist to cell specifically manipulate these proteins and establish the site of action.

The model of the mechanism of action of Rac1 and Arp2/3 has been modified (Figure 3G) to reflect this criticism. We have also modified the Discussion to acknowledge that these drugs may not only be acting presynaptically. While these proteins have been well studied postsynaptically, they also have established roles in presynaptic function (Gokhale et al., 2016; Njoo et al., 2015; Roesler et al., 2019). Given that our induction protocol relies only on presynaptic receptor activation (e.g. CB_1_ receptors are expressed at GABAergic terminals but not CA1 pyramidal neurons), which triggers a presynaptic structural change that relies on actin dynamics, a presynaptic target is the most likely explanation for our findings.

Reviewer #1:[…]Figure 1:While the detailed description of the analysis of the boutons was appreciated, the data presented in 1C are not convincing in their present state. The authors provide no controls to ensure that it's not endocytosis/remodeling of the CB_1_ receptors themselves, which would also result in a decreased CB_1_ volume. Additional inhibitory presynaptic markers (such as VGAT) should also be assessed and scored. Given the strict cutoffs the authors use in the analysis (ie sphericity value within the Imaris analysis), it could be that they're simply detecting a subset of smaller CB_1_ puncta resulting from local changes in the receptor pool; while the boutons themselves remain largely unchanged. The use of this CB_1_ volumetric analysis through the paper is problematic, as in almost all cases the authors do not test the effects of the pharmacological treatment alone (CHX, JSK, CK-666, NSC and ziram). Most if not all of the conclusions could be substantially undercut if these treatments in control conditions show the same change as when paired with WIN treatment. The study of structural plasticity in the postsynapse has largely relied on live cell imaging, so the compartment can be analyzed before, during and post treatment. While challenging, the authors should consider a similar approach, even if only to demonstrate their CB_1_ volumetric approach is valid. Using either an inhibitory neuron restricted GFP expression or expression of a tagged inhibitory presynaptic protein to allow live imaging of the bouton volume before and after WIN application, the authors could the perform fixation and staining for CB_1_ receptors and analyze how this compartment changed during the course of the treatments. This would bring their approach more in line with the standard in the field and allow for much more current and compelling analysis for structural plasticity to be performed.

We thank the reviewer for her constructive criticism.

Protein synthesis inhibition blocks the change in bouton volume, whereas receptor internalization or dimerization is not dependent on protein synthesis. The parameters used in Imaris include a lower size limit which was meant to exclude small CB_1_ puncta. Our experiment measuring Bassoon labeling within CB_1_+ boutons was designed as a control that is independent of changes in CB_1_. We have now also added data demonstrating that VGAT labeling is also reduced by CB_1_-iLTD induction (Figure 1—figure supplement 1E).

In response to concerns about internalization, we have performed an experiment to compare the subpopulation of CB_1_ receptors that colocalize with endosomal marker Rab7, following CB_1_-iLTD induction which showed no increase in CB_1_ internalization with WIN.

**Author response image 1. sa2fig1:** Left, Airyscan confocal images of Rab7/CB_1_ labeling. Right, % overlap of CB_1_ and Rab7 within CB_1_+ boutons was not significantly altered by 25 min WIN treatment when quantified as average per field of view nor by animal. CTRL: 0.09 ± 0.01 v. WIN : 0.06 ± 0.004, U = 18111; Z = 0.746, n.s. indicates p > 0.05; Mann-Whitney. n = number of fields of view (3 images/ slice, 1 slices/rat, 5 rats/CTRL, 4 rats/WIN). Data are presented as box plots (left) and data points (right) where box represents the 25th and 75th percentile of data range, mean is represented with a square, and median with a line inside the box.

In addition, we have added an in-depth discussion wherein we argue that it is very unlikely that receptor internalization and degradation could account for our findings.

We performed all drug only controls with JSK, CHX, and ziram alone and found they have no effect on basal bouton volume.

We attempted using CCK-Cre mice to test the viability of a Cre-dependent viral strategy to label CB_1_ interneurons with GFP, but we found this line to be leaky in excitatory neurons and we struggled to resolve boutons clearly given the background in the acute slices with 2-photon laser microscopy. We then attempted live imaging strategies with this interneuron population, by patch-loading CB_1_+ interneurons with morphological dyes as we did previously (Younts et al., 2016). Unfortunately, in our hands, the tiny caliber of the axon restricts dye loading and is difficult to resolve with the 2P to the extent needed to detect small changes in bouton volume.

Specific comments:– Examples (similar as in 1A) should be provided of CB_1_ boutons with WIN/WIN+CHX treatment.

We have now added representative boutons for WIN and WIN + CHX and additional widefield images in supplemental information to illustrate the boutons selected.

– CHX treatment alone should be assessed for C.

See Figure 1—figure supplement 1A

– CHX and WIN+CHX should be assessed for D.

See new Figure 1F

– Did the authors look at what happens to the corresponding inhibitory postsynaptic compartments? Presumably these also may decrease in size as well.

This is a very interesting question. Given that there are multiple reports of concurrent presynaptic changes resulting from postsynaptically-expressed forms of plasticity, we agree that there may be postsynaptic changes resulting from presynaptic plasticity. Testing this possibility is not straightforward given that inhibitory synapses mainly occur directly on the dendritic shaft, or in this case, on the cell soma, which makes structural analysis very difficult. In any event, we feel this issue is beyond the scope of our paper.

– For Figure 1—figure supplement 1C, the indicated mean for the WIN group for the graph provided should be checked. Given the large increase in the low volume population seen in the WIN group vs control group; and the similar size of the groups (540 vs 602) it's hard to understand how both means can be right around 1.

We have now added an additional technical replicate to this dataset (Figure 1—figure supplement 1C). Even with the addition of more “n”, there is still no significant difference between control and WIN-treated slices. We admit that while it appeared that the distributions were different in the previous dataset, we carefully rechecked the graph, and the means were accurate. We think that the population of larger boutons was also increased in the WIN group, but this was more difficult to detect by eye.

Figure 2:The experimental design (multiplexing of fully medium-heavy and heavy labeled cell cultures) is nice, but there are several potential issues regarding data processing and interpretation.Protein identification and quantification– The accepted parent mass tolerance of 50 PPM is very high and might lead to false identifications. When acquiring the data with an MS1 resolution of 60k on a well-calibrated Orbitrap LTQ instrument, a MS1 mass tolerance of ~5 PPM should be used.

The instrument error during acquisition was ± 15 PPM and was collected prior to calibrating the mass spectrometer. Though the precursor tolerance was ± 50 PPM the false discovery rate was adjusted to below 1% at the protein and peptide levels. The MS/MS of a large sampling of proteins were validated manually as well.

– Why are the SILAC labels (Lys4/Arg6 and Lys8/Arg10) specified as variable modifications instead of “SILAC pairs”? Please describe in more detail how the medium-heavy and heavy counterparts were quantified and how the fold-changes were calculated. Were only peptides species considered that were quantified in the medium heavy and heavy form? Why not using software that provide well-established workflows for SILAC quantification (such as the freeware MaxQuant)?

In Mascot the Quantitation Method (SILAC K+4 K+8 R+6 R+10) was used with each SILAC modification in exclusive mode. The raw data files were first processed using precursor ion quantitation of the Quantitation toolbox of Mascot Distiller (Matrix Science Ltd; version 2.7). Mascot was then used to search the rat databases (SwissProt and NCBI nr along with a decoy database to obtain FDRs) using the following parameters: trypsin; product ion mass tolerance of 0.40 Da; precursor ion tolerance of 50 PPM; carbamidomethyl Cys – fixed modification and variable modifications of: deamidated Asn and Gln; label:2H(4) of Lys; label:13C(6) of Arg; label:13C(6)15N(2) of Lys; label:13C(6)15N(4) of Arg and oxidation of Met. The result files obtained from Distiller and the Mascot searches were then uploaded to Scaffold Q+S (Proteome Software Inc.; version 4.9) using between-subjects, log_2_ ratio-based analysis of unique peptides against a reference to obtain the protein’s mean quantitative values and t-test p-values with Benjamini-Hochberg correction.

– From Supplementary file 1, we understand that each ~700 proteins were identified in the forward and the reverse samples. The overlap between the two samples is 391 proteins (main text), indicating that the reproducibility is fairly low between the measurements.Statistics

We thank the reviewer for pointing out the inconsistency. After careful re-examination of our results, we identified a misprint in the initial version of the manuscript that has now been corrected. The quantitation software Scaffold gives the Mascot search results showing 68% overlap (see Venn diagram in Author response image 2) which is quite good for MS/MS: 754 Proteins at 99.0% Minimum, 0.1% Decoy FDR with 2 unique peptides; 84,231 Spectra at 95.0% Minimum and 0.35% Decoy FDR. We have also updated Supplementary file 1 and Figure 2—figure supplement 1C accordingly.

– Please perform proper statistics to identify differentially regulated proteins (e.g. t-test and multiple testing correction). A simple fold-change cut-off (as described in Figure 2C) is not state-of-the-art.

We have now performed a new analysis of the dataset using statistical testing and multiple test correction (Benjamini-Hochberg) following the reviewer’s recommendation. We identified a total of 99 differentially expressed proteins (43 upregulated and 56 downregulated). Results of the analysis have been included in a volcano plot in Figure 2D as well as in Supplementary file 1.

Specific comments:– The supplementary data should be improved to enable the reader to understand the proteomics data and inspect all quantified proteins. Please include one table that presents the quantitative results for all proteins in all biological replicates (forward and reverse), so that the reader can get a sense for the biological variability. Please include a column with gene names. In Supplementary file 1, why does the table contain proteins from *rattus norvegicus*, *mus musculus* and human?

We have now edited the Supplementary file 1 following the reviewer’s request to include values for all biological replicates, gene name. The initial table had not been properly filtered by species. However, all analysis was conducted on filtered data from exclusively *Rattus norvegicus* proteins, and the table has been updated to reflect this.

– All mass spec raw data and database search results should be uploaded to the PRIDE repository prior to publication and the PRIDE identifier should be included in the Materials and methods section.

The mass spectrometry proteomics data have been deposited to the ProteomeX change with identifier Consortium via the PRIDE [1] partner repository with the dataset identifier PXD020008 and 10.6019/PXD020008.

Data interpretation:– Protein up-regulation is often interpreted as increased protein synthesis and protein down-regulation is often interpreted as increased degradation in the manuscript. However, based on the SILAC-MS data one cannot draw these conclusions. Protein up-regulation could result from more synthesis or less degradation and protein down-regulation could result from less synthesis or more degradation. Based on the presented data there is no evidence for one or the other.

The reviewer is correct that changes in protein levels could result from more or less synthesis/degradation and that these cannot be disentangled at the level of the SILAC-MS. We have now edited the manuscript to reflect this point. We do, however, provide evidence that ubiquitin-proteasome system activity and the degradation of ubiquitinated proteins is enhanced by CB_1_ activation in hippocampal slices (Figure 5A). Moreover, cycloheximide treatment blocks the reduction in bouton volume suggesting that protein synthesis is required.

– Based on the proteomics data one cannot distinguish proteins from different subcellular compartment or different cell types. The following statement from the Discussion sections is hence misleading: "[…] why ribosomal proteins would be synthesized locally if ribosomes are exclusively assembled in the nucleolus is unclear."

We have now removed this statement so as not to mislead the reader. We also clearly state in the Discussion that our results reflect protein isolated from whole neurons.

Figure 3:The authors refer to JSK as an "actin stabilizing drug", whereas in fact it's a drug that promotes actin polymerization. This fact complicates the data presented in this figure as essential controls (such as JSK treatment alone for 3A) are missing. If JSK treatment alone increases the bouton size, is this really demonstrating that the structural remodeling requires actin depolymerization?

New results indicate that JSK alone had no effect on the structure of boutons on its own (Figure 3A) suggesting that the blockade of structural change we see with WIN + JSK is due to the drug preventing an *activity-dependent* remodeling process. While some in vitro data supports the notion that JSK promotes actin nucleation, the effect of JSK is highly dependent on the presence of monomeric actin, other actin binding proteins, and the concentration and duration of JSK exposure (Bubb et al., 2000). There is also evidence for a paradoxical decrease in the amount of F-actin with JSK in some parts of the cell wherein actin remodeling was prevented via a sequestration of monomeric actin (Bubb et al., 2000; Spector et al., 1999). Overall, the net effect of JSK may be hard to predict, but all reports agree that it interferes with the normal polymerization of actin. For this reason and since we cannot measure exact F- or G-actin levels at these particular boutons, we have modified our conclusions with JSK accordingly to indicate only that CB_1_-iLTD relies on “actin dynamics” and not polymerization specifically.

Specific comments:– 3A, the examples provided with the black background are harder to examine. The authors should use the white background used in Figure 1 and Figure 1—figure supplement 1 examples.

Good suggestion; thank you. We have now made these changes.

– 3B, what explanation do the authors have for why the +JSK/WIN treated curve stabilizes around 80%?

We do not know but can speculate on two possibilities. The most likely explanation is that the mechanism of iLTD may rely only partially on the stabilization of filamentous actin. This is somewhat borne out by the structural data which is also not completely restored to baseline and may be due to the additional requirement of Arp2/3 or other actin binding proteins in iLTD. Alternatively, this could be a result of incomplete stabilization of actin polymers due to the inability of the drug to permeate deep enough into the tissue or other changes associated with JSK treatment such as sequestration of monomeric actin (discussed in more detail above).

– Additionally, the authors should test the effect of the Arp2/3 inhibitor and Rac inhibitor with and without WIN treatment on the bouton volume.

We attempted to perform this experiment but were interrupted by the COVID-19-related shutdown of the lab which resulted in the loss of the samples we had collected. Repeating these experiments involves a significant investment of time that is current magnified by the fact that our animal colonies, facilities, and laboratory access are all still operating at reduced capacity due to COVID-19 precautionary measures. However, given the strong correlation that is observed with all structural and functional manipulations (i.e. CHX, JSK, and ziram), our electrophysiology results with the Rac1 and Arp2/3 inhibitors strongly suggest concomitant structural changes or blockade thereof.

Figure 5:Specific comments:– 5D, examples would be better with white backgrounds.

We have now changed the representative boutons accordingly.

– Analysis for the effect of ziram alone needs to be done.

Agreed; please, see new Figure 5D.

Reviewer #2:[…]1) Boutons simply should not be considered individual n's for statistics. The authors are welcome to argue for what the independent biological unit is. Please also clarify, for a given condition (such as Veh or WIN), were slices from the same animal used for multiple conditions, or were the slices from one animal used for one condition at a time? If the latter, then an animal is the biological unit, with the possibility of using nested data statistical approaches. Alternatively, the mean value/slice could be considered. Issues such as animal health, temperature fluctuations, buffer osmolarity et cetera could otherwise exist and generate 100s of values per slice that are a little bigger or smaller due to technical variables. This is a foundational observation for the study and needs to be robust. As presented, it looks small and only visible when n's are in the 1000s. Likewise, although it was good to consider involvement of other inhibitory synapses (like PV^+^), a figure showing no significant difference that is potentially grossly underpowered is not so useful (n's of 500).

In these figures, we have used slices from the same rats for multiple conditions. We absolutely agree that slices are vulnerable to factors such as temperature fluctuations, changes in buffer osmolarity/oxygenation, etc. For this reason, each experiment and all conditions were performed at the same time on the same day with the same buffer using slices from the same animal. In order to further balance the conditions, slices were matched by location in the dorsalventral axis across the conditions. We also image at 3 specific locations in the CA1 pyramidal layer to tile the area between CA2 and subiculum. However, even with all these variables controlled for, it is still highly expected that boutons will not be uniform in size. We feel that the intrinsic variability in volume present in the population of boutons is biologically relevant and should be represented for the reader as has been previously reported (Dudok et al., 2015; Lees et al., 2019; Meyer et al., 2014). Moreover, while we fully acknowledge that the percent change we see in bouton volume is small, it is clear that small changes in the nervous system are no less functionally important. According to our power analysis, in order to achieve 80% power, given the standard deviation of bouton volume per slice was averaged at ~0.45, we would require 142 slices per group for an effect size of 10% (see Author response image 3). Such a high number would be next to impossible to achieve with any standard animal protocol.

**Author response image 3. sa2fig3:** 

As for the analysis of inhibitory synapses established by PV^+^ neurons, we would anticipate a similar “n” is necessary to see a WIN-induced change and therefore n’s of 500 are well-within range. However, to make the power of the PV control experiment more comparable to those performed on CB_1_+ boutons we have now added additional “n” to this experiment. We still see the WIN treatment has no effect on PV bouton volume. We also should note that although all datasets presented in this paper represent multiple technical replicates pooled together, we routinely see the WIN induced shrinkage in each individual technical replicate with much lower ‘n’s and the WIN effect is evident if we average the bouton volume values from Figure 1C per slice (see Author response image 4).

**Author response image 4. sa2fig4:** 

2) In general, the bioinformatics approach was rigorously and thoroughly presented. It is also nice that the authors further discuss results from other programs. One item to clarify – since the GO analyses are the major data presented: please review and clarify the extent to which the same protein is annotated with multiple functional properties such that it is populating (and driving) multiple terms on those lists.

We agree with the reviewer regarding the annotation redundancy in Gene Ontology (GO) analysis. In order to overcome that issue, we have attempted a two-pronged approach: (i) we used several annotation tools (GSEA, IPA and SynGO) and (ii) we performed Enrichment map analysis of GSEA (Figure 2C). Here, each node represents a GO term and edges represent overlap between gene sets. This analysis highlighted 4 dense clusters (e.g. protein synthesis and processing, neuronal projections, energy metabolism and extracellular space). The observation of distinct clusters of enriched GO terms demonstrates the low overlap between gene-sets and confirms that the associated functional properties are populated by distinct gene-sets. Within a given cluster, there is indeed a higher level of similarity as indicated by the proximity between the nodes (see also Supplementary file 2).

Reviewer #3:[…]The manuscript contains an interesting data set and the involvement of the ubiquitination system in CB_1_-LTD is highly intriguing. However, after reading the manuscript there are many “lose ends”, and it is still not clear what happens during CB_1_-LTD. Indeed, the authors themselves describe these events as “coordinated engagement of multiple cellular processes”, which does not clarify much. I would like to encourage the authors to try and formulate a more complete hypothesis of the cellular events that underlie CB_1_-LTD.

We thank the reviewer for their comments about our study. To clarify our hypothesis of the molecular events that underlie CB_1_-LTD, we have now modified the text and summarized the molecular mechanisms underlying CB_1_-iLTD in a diagram (Figure 5E). Our SILAC data clearly show an increase in several proteins that are involved in proteasomal function, raising the possibility that protein synthesis is upstream of protein degradation, and not in parallel.

I have two major points:1) It is not easy to link the proteomics findings to changes in the actin cytoskeleton in presynaptic boutons. For instance, the authors show that WIN slightly reduces Arp2c protein levels, but that inactivating Arp2/3 actually enhances CB_1_-LTD. I am wondering if the explanation given by the authors the most plausible interpretation of the data. Is it known how CB_1_-LTD is expressed (fewer vesicles? reduced release probability?) and is it possible to link this to specific changes in the actin cytoskeleton?

CB_1_-iLTD causes a reduction in neurotransmitter release, i.e. reduced release probability (Chevaleyre and Castillo, 2004; Heifets et al., 2008), however the mechanism(s) driving this phenomenon are, as yet, unknown. We have attempted to address this question here, but more work is still needed to understand the exact molecular pathways involved. Early research suggested the mechanism of CB_1_-iLTD might rely on dephosphorylation of the active zone protein RIM1α (Chevaleyre et al., 2007), but phosphorylation of RIM1α by PKA was later shown not to be involved (Kaeser et al., 2008). CB_1_ receptor activation has also been suggested to control release via change in vesicular release mode (Aubrey et al., 2017) or change in the distribution of vesicles around the active zone (Garcia-Morales et al., 2015; Ramirez-Franco et al., 2014), but these studies were performed at different synapses that may not express the same form of CB_1_-iLTD. In general, it is well-established that the size of the presynaptic bouton and active zone is correlated with the postsynaptic response (Bartol et al., 2015; Bourne et al., 2013; Meyer et al., 2014). Here, we suggest that a smaller bouton and active zone in combination with ubiquitination and subsequent degradation of presynaptic release machinery proteins could explain the reduction in GABA release associated with iLTD.

The finding that Arp2/3 inactivation enhances CB_1_-iLTD may seem a bit counterintuitive but in fact fits well with literature demonstrating that loss of interaction between WASF1/WAVE1 and Arp2/3 causes a loss of functional presynapses (Wegner et al., 2008) and CB_1_ activation drives growth cone collapse in a Rac1 dependent manner (Njoo et al., 2015) Our SILAC and WB data show that CB_1_-iLTD triggers loss of ARPC2 and WASF1/WAVE1 which functionally would have the effect of reducing neurotransmitter release (i.e. LTD). Presumably, when we apply CK-666 we are further reducing the pool of Arp2/3 available to participate in the maintenance of actin branches, on top of the Arp2/3 protein that is presumably being degraded, and that is why we see enhanced LTD (i.e. further reduction in neurotransmitter release).

2) The proteomics data show that multiple processes are activated during CB_1_-LTD. The manuscript does not provide a model or hypothesis for how these processes are linked. For instance, are ubiquitination and protein synthesis independently regulated by CB_1_? What is the link between ubiquitination and presynaptic actin remodeling? Would it be possible to link CB_1_-LTD to a specific E2 or E3 ligase in the presynapse?

We have now provided a proposed model in Figure 5E. Based on two pieces of data, the most likely scenario seems to be that protein synthesis triggered by CB_1_ activation via mTOR (Younts et al., 2016) is upstream of ubiquitination and structural change. CB_1_ activation triggers an upregulation of several proteasomal subunits and E2 ligases (Figure 2, 4), which suggests these components may be synthesized on demand to regulate protein degradation locally. While we cannot exclude that their upregulation comes from a reduction in their degradation, the long half-life of proteasomal subunits makes it unlikely that we could see such a strong increase from reduction in degradation alone in such a short time window (25 min)(Dorrbaum et al., 2018). Structural change requires protein synthesis. Our data suggests that CB_1_ activation induces rapid reduction of ARPC2, WASF1/WAVE1 and other components of the WAVE regulatory complex (WRC), and that this pathway is required for iLTD. We propose that ubiquitination of these proteins underlies structural remodeling.

[Editors' note: further revisions were suggested prior to acceptance, as described below.]

Requested revisions:1) The authors in their rebuttal argue that standard power calculations are "unreasonable" because n of 145 slices would be required. At the most fundamental level, if this were true, this question and conclusion are not appropriate given the constraints of the methodology and one should focus on powerable questions. However, a statistical analysis by a statistician might indicate other analytical approaches. The authors show how close to being powered they may be with the slice means alone (not including nesting) showing significance. If the authors want to show the data as is in the main figure, the authors could address this concern by creating supplemental figures that show bouton size distributions for each slice and also figures with slice means and exact p values.

Excellent point. In response, we have created new supplemental figures (Superplots) that show the bouton size distribution color-coded by slice with the slice means plotted on top and the overall mean of slices overlaid with a vertical line representing the 95% confidence interval (CI) to clearly represent the effect size (Lord et al., 2020). For these figures, we reanalyzed the data and balanced the number of slices/animals across conditions and performed one-way ANOVA with post hoc Tukey test for multiple comparisons. Exact p-values and confidence intervals are included in each supplemental figure legend. Importantly, our key observations were reproduced using slice as the independent variable.

2) Bassoon puncta were to serve as an independent measure of bouton size to support the CB_1_ observations. However, the methods describe first making a binary mask of CB_1_ signal, then using the "and" function to isolate the bassoon pixels that overlap with CB_1_ pixels. As described, it is almost impossible to reach any other conclusion than reproducing the CB_1_ measurement results. Please clarify methods substantially to remove this concern.

We apologize for the confusion. A dilated CB_1_ mask was used to isolate Bassoon in the vicinity of and inside CB_1_ boutons, the size of Bassoon is free to vary inside the presynaptic terminal since Bassoon labeling is considerably smaller than the CB_1_ terminal volume (Average control volume of bassoon: ~ 0.51 μm^3^ v. CB_1_: 1.85 μm^3^). We acknowledge the limitations of our method and have removed mentions of the CB_1_-independent nature of this measurement. Of note, our findings are consistent with a large body of previous work showing the active zone size is highly correlated to the overall presynaptic size (Matz et al., 2010; Maus et al., 2020; Meyer et al., 2014).

3) In working to propose a solution for data presentation, the authors raise an important point, supplemental data should include an analysis that addresses whether the plasticity effects preferentially involve a particular size subset, smaller or large.

We have now included a histogram showing binned bouton distribution per slice to address this question (Figure 1—figure supplement 1C). These data suggest that iLTD results in a reduction in the number of large boutons and an increase in the fraction of small boutons.

[Editors' note: further revisions were suggested prior to acceptance, as described below.]

Reviewer #2:1) The supplementary figures are highly appreciated and allow critical review of the nature of the experimental data and robustness of the conclusions.2) The authors' response indicating that data was removed to balance slice numbers needs to be clarified. One should not have to remove data. If slices were removed from supplementary figure analyses, what guided those decisions? Do the removed slices remain in the main figure data? I would suggest all the data that were believed to be valid in the original submission be included in the main and supplementary figures. (Or was there a situation of grossly unequal n's in the control/win condition and the authors' are trying to make statistical comparisons across similarly sized groups post hoc? In principle, a similar number of interleaved controls should exist for each additional test condition.) Please clarify.

We have made the requested clarification points in the manuscript. The reviewers suggested we include *all* slices and use “slice” as our independent variable. Following their advice, we went ahead and included plots showing average bouton volume per slices in the main figures (similar to the plots we included in our response to the reviewers) and kept the superplots in supplementary figures.

**References**

Bubb, M.R., Spector, I., Beyer, B.B., and Fosen, K.M. (2000). Effects of jasplakinolide on the kinetics of actin polymerization. An explanation for certain in vivo observations. J Biol Chem *275*, 5163-5170.

Dorrbaum, A.R., Kochen, L., Langer, J.D., and Schuman, E.M. (2018). Local and global

Feng, Y., Press, B., and Wandinger-Ness, A. (1995). Rab 7: an important regulator of late endocytic membrane traffic. J Cell Biol *131*, 1435-1452.

Gokhale, A., Hartwig, C., Freeman, A.H., Das, R., Zlatic, S.A., Vistein, R., Burch, A., Carrot, G., Lewis, A.F., Nelms, S.*, et al.* (2016). The Proteome of BLOC-1 Genetic Defects Identifies the Arp2/3 Actin Polymerization Complex to Function Downstream of the Schizophrenia Susceptibility Factor Dysbindin at the Synapse. J Neurosci *36*, 12393-12411.

Howlett, A.C., Blume, L.C., and Dalton, G.D. (2010). CB(1) cannabinoid receptors and their associated proteins. Curr Med Chem *17*, 1382-1393.

Kaeser, P.S., Kwon, H.B., Blundell, J., Chevaleyre, V., Morishita, W., Malenka, R.C., Powell, C.M., Castillo, P.E., and Sudhof, T.C. (2008). RIM1alpha phosphorylation at serine-413 by protein kinase A is not required for presynaptic long-term plasticity or learning. Proc Natl Acad Sci U S A *105*, 14680-14685.

Lees, R.M., Johnson, J.D., and Ashby, M.C. (2019). Presynaptic Boutons That Contain Mitochondria Are More Stable. Front Synaptic Neurosci *11*, 37.

Maus, L., Lee, C., Altas, B., Sertel, S.M., Weyand, K., Rizzoli, S.O., Rhee, J., Brose, N., Imig, C., and Cooper, B.H. (2020). Ultrastructural Correlates of Presynaptic Functional Heterogeneity in Hippocampal Synapses. Cell Rep *30*, 3632-3643 e3638.

Roesler, M.K., Lombino, F.L., Freitag, S., Schweizer, M., Hermans-Borgmeyer, I., Schwarz, J.R., Kneussel, M., and Wagner, W. (2019). Myosin XVI Regulates Actin Cytoskeleton Dynamics in Dendritic Spines of Purkinje Cells and Affects Presynaptic Organization. Front Cell Neurosci *13*, 330.

Spector, I., Braet, F., Shochet, N.R., and Bubb, M.R. (1999). New anti-actin drugs in the study of the organization and function of the actin cytoskeleton. Microsc Res Tech *47*, 18-37.

Wegner, A.M., Nebhan, C.A., Hu, L., Majumdar, D., Meier, K.M., Weaver, A.M., and Webb, D.J. (2008). N-wasp and the arp2/3 complex are critical regulators of actin in the development of dendritic spines and synapses. J Biol Chem *283*, 15912-15920.